# Seismicity at the Castor gas reservoir driven by pore pressure diffusion and asperities loading

Simone Cesca [1✉], Daniel Stich[2,3], Francesco Grigoli [4,5], Alessandro Vuan [6],
José Ángel López-Comino [2,3,7], Peter Niemz [1,7], Estefanía Blanch [8,9], Torsten Dahm [1] &
William L. Ellsworth [10]

The 2013 seismic sequence at the Castor injection platform offshore Spain, including three earthquakes of magnitude 4.1, occurred during the initial filling of a planned Underground Gas Storage facility. The Castor sequence is one of the most important cases of induced seismicity in Europe and a rare example of seismicity induced by gas injection into a depleted oil field. Here we use advanced seismological techniques applied to an enhanced waveform dataset, to resolve the geometry of the faults, develop a greatly enlarged seismicity catalog and record details of the rupture kinematics. The sequence occurred by progressive fault failure and unlocking, with seismicity initially migrating away from the injection points, triggered by pore pressure diffusion, and then back again, breaking larger asperities loaded to higher stress and producing the largest earthquakes. Seismicity occurred almost exclusively on a secondary fault, located below the reservoir, dipping opposite from the reservoir bounding fault.

[1] GFZ German Research Centre for Geosciences Potsdam, Potsdam, Germany. [2] Instituto Andaluz de Geofísica, Universidad de Granada, Granada, Spain. [3] Departamento de Física Teórica y del Cosmos, Universidad de Granada, Granada, Spain. [4] Department of Earth Sciences, University of Pisa, Pisa, Italy. [5] Swiss Seismological Service, ETH Zurich, Zurich, Switzerland. [6] National Institute of Oceanography and Applied Geophysics - OGS, Trieste, Italy. [7] Institute of Geosciences, University of Potsdam, Potsdam-Golm, Germany. [8] Departament de Física-EPSEB, UPC Barcelona Tech, Barcelona, Spain. [9] Observatori de l'Ebre (OE), CSIC—Universitat Ramon Llull, Roquetes, Spain. [10] Department of Geophysics, Stanford University, Stanford, CA, USA. ✉email: simone.cesca@gfz-potsdam.de

Anthropogenic seismicity is stimulated by stress perturbations, fracturing processes, or pore pressure changes in the subsurface accompanying fluid and mass movements driven by industrial activities[1,2]. Since these human actions only affect the uppermost several kilometers of crust, induced earthquakes are expected to occur at similar depths. As a matter of fact, almost all major cases of injection-related induced seismicity with an accurate hypocentral location occurred at ~4–8 km depth (Supplementary Table 1). Due to its shallow depth, induced seismicity poses an important seismic hazard, as even moderate magnitude earthquakes can produce strong shaking and localized damages[3–6]. For these reasons, anthropogenic seismicity has become a matter of great societal concern and represents a new focus for seismic hazard analysis[1,7–9]. While induced seismicity accompanied the exploitation of different natural resources over the past century[7,10–14], so-called injection-induced seismicity (IIS) has gained scientific attention in the past few decades[1,15,16]. The interest in IIS has been prompted by the recent occurrence of a number of important earthquakes, which have been associated with fluid injection in the frame of hydraulic fracturing[17–19], wastewater disposal[4,20], conventional hydrocarbon extraction[7,14], and geothermal exploitation operations[5,6,21].

Underground gas storage (UGS) facilities permit storage and extraction of large volumes of natural gas to manage fluctuations in demand or provide a strategic reserve. They are operated routinely at hundreds of sites globally, typically using depleted oil/gas reservoirs, aquifers, or caverns in rock/salt formations. This type of injection/extraction operations typically has a weak seismic impact, and only a few cases of seismicity potentially related to gas storage have been reported. Gas storage-induced seismicity may have occurred at the Gazli gas storage, Uzbekistan, after the previous depletion of a gas field, which had already experienced larger damaging earthquakes in 1976 and 1984[22]. When the reservoir was used for gas storage, cycling injection/extraction operations were accompanied by earthquakes up to M 5[23]. Another case of gas storage-induced seismicity has been proposed for the Hutubi UGS, Xinjiang, China[24], where the largest earthquake reached Mw 3.0 and was located at ~4 km depth, slightly deeper than the reservoir formation. More recent cases of induced microseismicity in Europe, reported at different gas storage facilities[2,25], did not exceed Mw 1.0. The general lack of induced seismicity observations at UGSs may be partially attributed to poor seismic monitoring[24], but even well-monitored gas reservoirs, such as the Collalto reservoir, NE Italy, showed an almost complete absence of induced seismicity, with no microseismicity above M 0.0 within 3 km from the reservoir[26]. Another reason for the low seismogenic potential of gas storage operations could be that gas injection schedules are typically designed and engineered not to exceed the stress conditions existing prior to or during the original reservoir production. To date, the most important case of seismicity induced by gas storage in Europe is the Castor project, offshore Spain, which has been considered to have triggered a seismic sequence of >1,000 earthquakes, peaking with three M 4+ earthquakes in September–October 2013[27,28].

At the Castor project, cushion gas was injected from a platform, located ~22 km offshore the coast of Spain, into a formerly depleted hydrocarbon reservoir (Fig. 1). The reservoir itself extends along the NNE–SSW direction and deepens eastward. The geology is characterized by a karsted, fractured limestone. The reservoir is sealed on the NW by the Amposta fault, a normal fault acting as the reservoir roof. The other sides of the reservoir are sealed by an aquifer[29]. When gas is injected, it remains trapped at the top of the reservoir, and as the injection continues, this will displace the gas–oil–water interfaces downwards.

A series of injection tests were performed at Castor starting in 2013[29]. Seismicity was first observed close to the Castor project platform on September 5, 2013, 3 days after the start of a large-scale gas injection in the depleted Amposta oil reservoir[27,29]. The close proximity of the hypocenters to the injection wells, the temporal correlation between injection operations and seismic unrest, and the low natural background seismicity in the epicentral region were a strong argument for early evidence of triggered or induced seismicity[27–32]. The seismicity increased in rate and maximum magnitude, reaching a magnitude MbLg 3.0 on September 13, 2010 (Instituto Geografico Nacional, IGN, catalog). Injection operations were stopped on September 17, 2013[29]. Nevertheless, as observed for other induced seismicity cases[4,5,33], seismicity continued in the following days. Larger earthquakes occurred, peaking in three earthquakes with magnitude Mw >4 in early October[27], before finally fading away. The Castor project was permanently closed in 2019.

Seismicity patterns and their relationship to gas injection operations have been discussed in many early publications and open reports on the Castor project, assessing epicentral locations[27–29], hypocentral depths[27–29,31,32], focal mechanisms and moment tensors[27,29,31], statistical seismicity parameters[27,30], the local velocity structure[28], and potential mechanisms for fault reactivation[27,31,32,34]. Unfortunately, the growing number of scientific publications has not been accompanied by a clearer understanding of the seismogenic processes at Castor, as some seismological results remain debated, and ultimately there is no common agreement on which fault(s) was activated. The first major question concerns the overall spatial distribution of the epicentral locations, potentially providing information on the orientation of activated fault(s). In one study, the orientation was found to be roughly parallel to the coast[27] and perpendicular to it in another[28]. As for the hypocentral depth, there is some rough agreement on the identification of shallow crustal sources but with a broad range of depth estimates, mostly varying between 1–5[27] and 6–8 km[28,32] but including even deeper estimates[28,29]. Such large uncertainty prevents development of reliable models for fault activation. Finally, there is consensus on the focal mechanism, showing strike-slip mechanisms with a steep NW–SE plane and a SE dipping NE–SW plane, with some variability on the dip angles[27,29,31]. Unfortunately, each of the possible two plane orientations fits one of the proposed seismicity distributions, so that the fault geometry remains unresolved. Bringing together previous results is challenging, given the variety of data, velocity models, and seismological methods and the lack, in most previous studies, of reliable uncertainty estimations.

Here we recollect a broad dataset from 31 seismic stations (Supplementary Fig. 1), including one ocean bottom seismometer (OBS), substantially increasing the dataset and/or the azimuthal coverage, in comparison to previous works, either using <10 stations[27,29] or <20 with a strongly asymmetric network geometry[31]. We model seismic data by a combination of modern seismological techniques, including a probabilistic moment tensor inversion, empirical Green's Function, far-regional and teleseismic array analysis, and template matching, which are used for the first time to analyze this seismic sequence. While the assessment of parameter uncertainties allows explaining previous partially contradictory results, our new results significantly improve the location accuracy and source parameter resolution, which allows us to reconstruct the complex, multi-stage evolution of seismicity and propose a self-consistent rupture scenario for the seismic unrest.

## Results

We adopt a range of modern techniques (see details in the "Methods" section), which provide substantial new results. Template matching is used to enhance the catalog size, relative

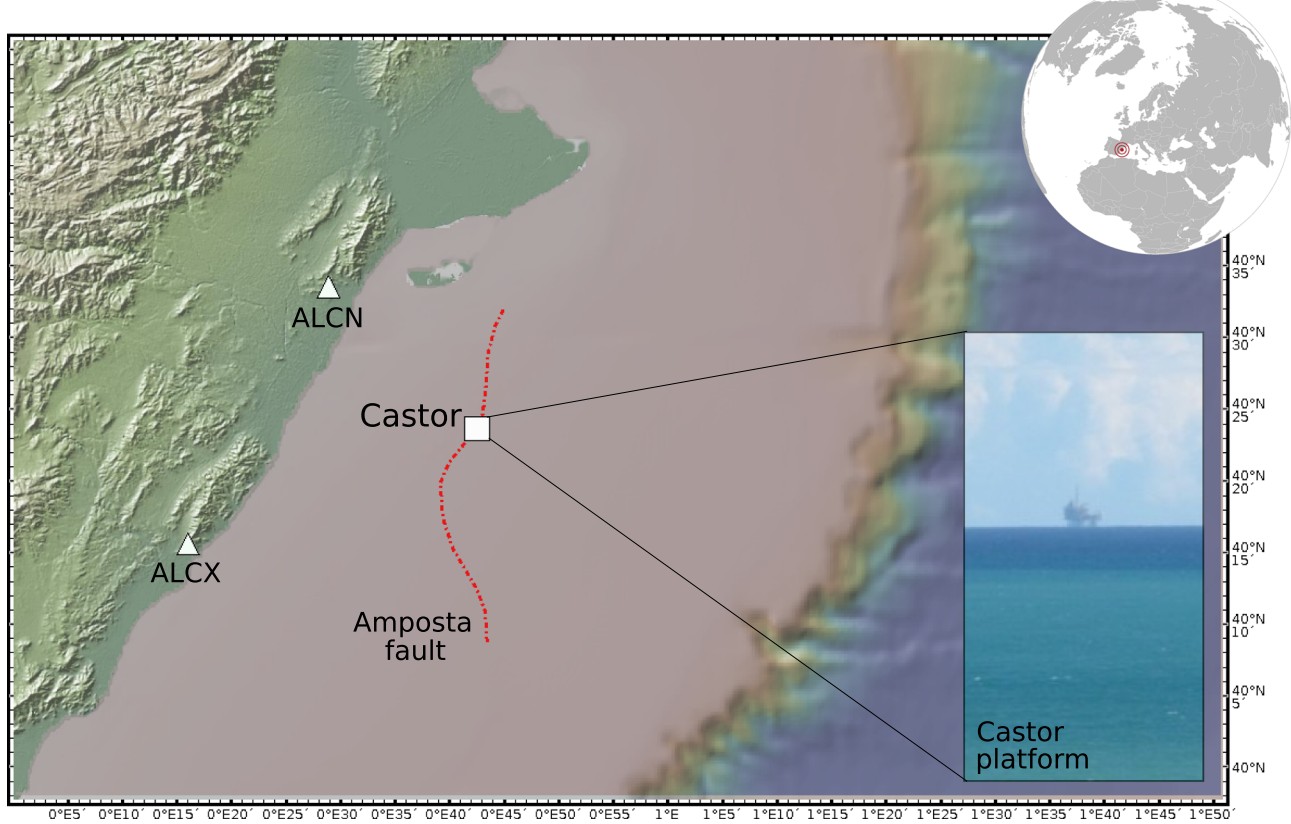

**Fig. 1 Overview map of the study region offshore Spain.** The map shows the location of the Castor platform (white square), closest seismic stations ALCN and ALCX (white triangles), and the approximate trace of the Amposta fault (red dotted lines)[47,48]. The upper right inset shows the geographical location of the study area, the lower right inset a photo of the Castor platform (courtesy Álvaro González). The map has been plotted with GeoMapApp (www.geomapapp.org), using the Global Multi-Resolution Topography (GMRT)[70] and regional bathymetry after the EMODNET 500 m compilation.

location methods to establish the controversial distribution of epicenters, array techniques to infer accurate depth estimates based on independent data, moment tensor and rupture directivity analysis to constrain the rupture plane orientations, and finally an accurate assessment of waveform similarity to confirm the overall similarity in location and focal mechanism. Results stability is assessed using the local velocity model G[28], testing alternative models (Supplementary Fig. 2), and providing source parameters uncertainties, which help to explain some inconsistencies among previous studies. The new results resolve details of the evolution of seismicity with ongoing injection and shed light on the physical causes of induced seismicity and its magnitude.

**Template matching detection.** We reprocess continuous recordings to enhance the seismic catalog with the aim of better tracking the seismicity evolution in space and time. Using waveform template matching (examples of three detections at different magnitude levels are shown in Supplementary Figs. 3–5), we are able to considerably augment the detected earthquakes from 536 (IGN catalog) and 982 (Ebro catalog) to 3437 earthquakes (Supplementary Fig. 6 and Supplementary Dataset 1). The augmented catalog includes the 148 templates used. The minimum magnitude in our extended catalog is Mw −0.2. The magnitude of completeness improved from Mc ~2.0 (IGN)[35] and Mc 1.3 (Ebro)[27] to Mc 0.6 and 0.3 in the injection and post-injection periods, respectively (Supplementary Fig. 7), using magnitudes of the Ebro catalog as reference. The analysis of the extended catalogs confirms a substantial change in the seismicity occurred on September 17, 2013, when injection operations were

interrupted (Supplementary Fig. 7). The b-value dropped from 0.99 ± 0.04 to 0.77 ± 0.01 between the injection and post-injection seismicity phases, supporting previous findings based on catalog alone[27].

**Earthquake relocation.** We use two independent, advanced relative earthquake localization methods, based on waveform cross-correlation and $t_S$-$t_P$ differential times combined with distance geometry techniques. Both methods take advantage of the P and S waveforms and have a higher resolution than the conventional, absolute localization methods used in previous studies for the Castor sequence[27–29]. These techniques are especially suited for the analysis of spatially clustered seismicity, as observed at Castor, and less sensitive than absolute location techniques to structural heterogeneities outside the seismogenic volume and to network asymmetry. Waveform-based relative hypocentre locations obtained for a subset of 51 events with magnitudes >2.0 (Supplementary Dataset 2) demonstrate that the sequence was restricted to a single, ~4 km long, N42°E trending lineament (Fig. 2a), even after accounting for uncertainties (Supplementary Fig. 8). Precise locations result from cross-correlation relative timing, and achieved an average root mean square travel time errors of 0.02 s. Hypocenters concentrate within an ~1 km wide depth interval (Fig. 2c, d), although relative depths are difficult to resolve due to the asymmetric network geometry (Supplementary Fig. 1). As discussed below, the absolute depth of seismicity is less well resolved and likely at a depth of ~3 km.

Locations developed using an independent and alternative approach[36] confirm the orientation of the seismicity. The method relies on the solution of a seismological distance geometry

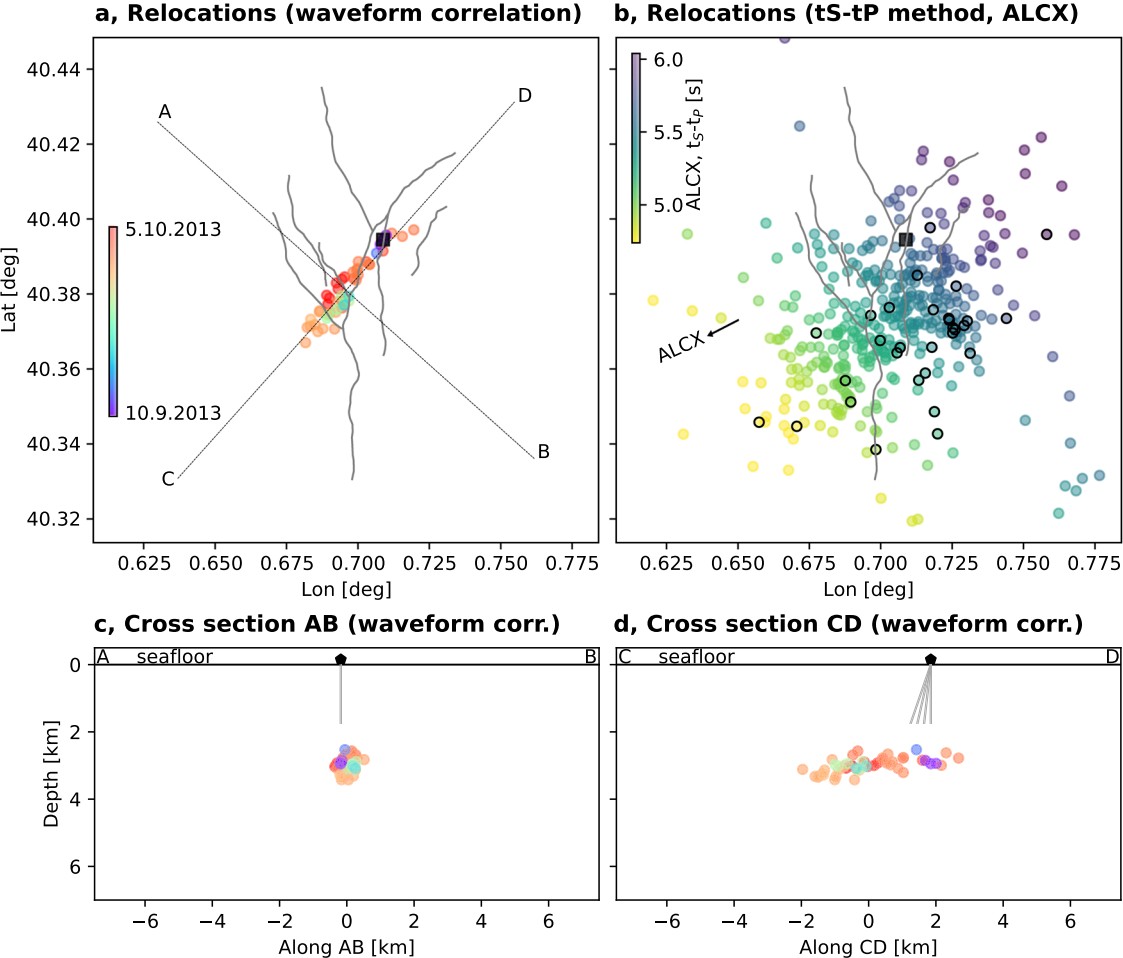

**Fig. 2 Locations based on waveform correlations and $t_S$-$t_P$ method. a, c, d** Locations based on waveform correlations in map view and cross-section. **b** Locations based on $t_S$-$t_P$ method in cross-section. Earthquake locations (circle, colors denoting origin time in **a**, **c**, **d**, and S-P differential time at station ALCX in **b**, according to color bars in **a** and **b**, respectively) are plotted in map view (**a**, **b**) and along AB (**c**) and CD (**d**) cross-sections (profiles AB and CD are shown in **a**). Black circles in **b** denote those events used in the waveform correlation location (**a**), for which S-P time estimates are available. A pentagon denotes the location of the Castor platform, double gray solid lines sketch the injection wells[29].

problem[36] using differential S-P travel times at the closest seismic stations, ALCN and ALCX. These two stations, the closest to the Castor platform, are optimally situated for this method, having almost perpendicular azimuths with respect to the seismicity (Fig. 2b and Supplementary Fig. 9). While more scattered, locations were obtained for a larger dataset of 408 earthquakes (Supplementary Dataset 3). The earthquakes located with the waveform correlation approach have a larger range of S-P differential times at ALCX (SW), than at ALCN (NW), which confirms the NE–SW trend of the seismicity (Supplementary Fig. 9). Location results are stable and not substantially biased by the assumed velocity model (Supplementary Fig. 10). Both catalogs are included in the Supplementary Material.

**Moment tensor inversion.** Moment tensor solutions of 11 earthquakes with Mw >3.0 (Supplementary Table 2) are based on a probabilistic inversion approach[37] and considers different one-dimensional (1D) velocity models and wavefield attributes (Supplementary Figs. 11–13). The results are stable and independent of the velocity models used. All studied events display a similar, predominantly strike-slip faulting style (Fig. 3). The left-lateral nodal plane trends NE–SW, roughly parallel to the coast, and dips toward SE (strike 42° ± 3°, dip 48° ± 11°, rake −1° ± 8°, estimated from the distribution of best quality A and B solutions

for 9 earthquakes assuming model G, Supplementary Table 2). The right-lateral nodal plane, trending NW–SE (strike 313° ± 4°), is sub-vertical. The enhanced catalog of moment tensors resembles the general focal mechanism proposed in previous works[27,28,31]. Centroid locations depend on the chosen model, with the average locations closest to the injection platform for model G. Centroid depths average 3.4 ± 1.6 km. The largest magnitudes of Mw 4.09 ± 0.04 are found for model G. Similar results are obtained for other velocity models, with a few cases having slightly greater depths and magnitudes (Supplementary Table 2).

**Focal depth estimation from depth phases at regional and teleseismic distances.** We conducted an independent analysis of the focal depth for the three largest events using depth phases observed at regional and teleseismic distances on different arrays, using the abedeto tool (https://github.com/HerrMuellerluedenscheid/abedeto). Green's functions computed using local velocity models at both the source and array locations (details in the Supplementary Note 8) form synthetic beams that also account for the moment tensors estimated in this study. Measured time delays between direct P and the near-surface reflected pP phase are of the order of 1.5–1.8 s, as estimated at the GERES array[38], Germany (~1400 km distance). We obtained similar results using the ILAR, US array at teleseismic

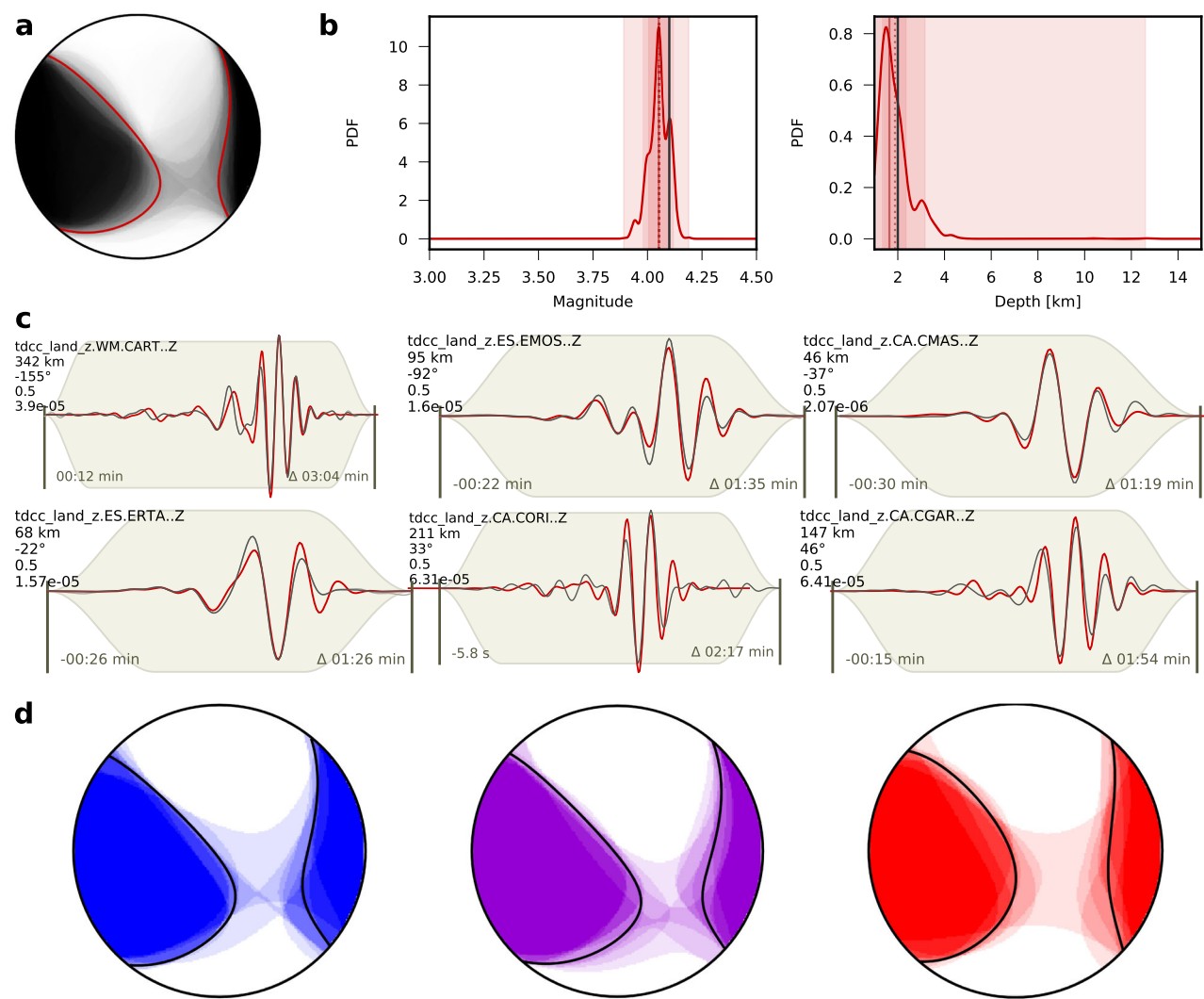

**Fig. 3 Overview of the moment tensor inversion results. a** Overlay of deviatoric moment tensor solutions obtained for the October 2, 2013, 23:06 UTC earthquake (Supplementary Table 2) out of the bootstrap approach (gray focal spheres) and best solution using all data (red thick line focal sphere). **b** Probability density functions for moment magnitude (Mw) and depth for the same event (red dotted and solid line indicate best and mean solutions, dark to light red background confidence levels of 68, 90, and 95%). **c** Examples of waveform fit (red and black lines denote synthetic and observed displacement waveforms, respectively) at selected stations for the same event (station name, azimuth, distance, weight, and maximal displacement are reported). **d** Overlay of moment tensor solutions for 9 studied events (quality A and B, Mw 3.4–4.1), assuming three different velocity models: model I (blue), G (purple), and V (red focal sphere); thick black lines denote the focal sphere of the cumulative moment tensor.

distances (~8000 km). These delays are compared to the moveout and waveform patterns synthesized from theoretical seismograms at each array location. Synthetic waveforms and delays are consistent with centroid depths of 3–4 km (Fig. 4 and Supplementary Fig. 14).

**Rupture directivity.** The rupture process for the two largest earthquakes of the sequence, the October 1, 2013, 03:32 Mw 4.1 (EQ1) and October 2, 2013, 23:06 Mw 4.1 (EQ2) earthquakes, retrieved by applying empirical Green's function (EGF) techniques[39], show similar apparent source time functions (ASTFs), with durations of 0.25–0.72 s (Fig. 5 and Supplementary Fig. 15). A clear azimuthal pattern of the ASTFs is identified for both events, from fitting S wave-based apparent durations (Fig. 5). The resulting source parameters (Supplementary Table 3) show that both earthquakes share similar rupture characteristics. We estimate rupture lengths of ~1.1 km, in agreement with the size appropriate for a crack model (see "Discussion"), with a rupture duration of ~0.5 s. The intrinsic trade-off between rise time and rupture velocity[40,41] leads to high uncertainties;

assuming a rise time of 0.15–0.25 s, the estimated rupture velocity would be ~2.7 km/s, similar to the shear wave velocity at 3 km depth in model G. Ruptures from both earthquakes are found to be asymmetric bilateral, with most of the rupture (66 and 75% of the overall rupture length for the two earthquakes, respectively) propagating toward NNE from the initial hypocenter. These results better correlate with the N42°E-oriented fault, dipping SW, which is resolved independently from the band of seismicity and the moment tensor solutions, and they are inconsistent with a rupture along the alternate plane of the moment tensor, which is sub-vertical and strikes to NW.

**Waveform-based classification.** We employ network-based event similarity clustering[42] to identify events with high waveform similarity at multiple stations, which implies high similarity in their locations and focal mechanisms. The subset of 51 relocated events, with magnitudes >2.0, shows very similar waveforms across the station network. Surface waves are highly similar but only visible for larger magnitude events, so we analyzed P and S

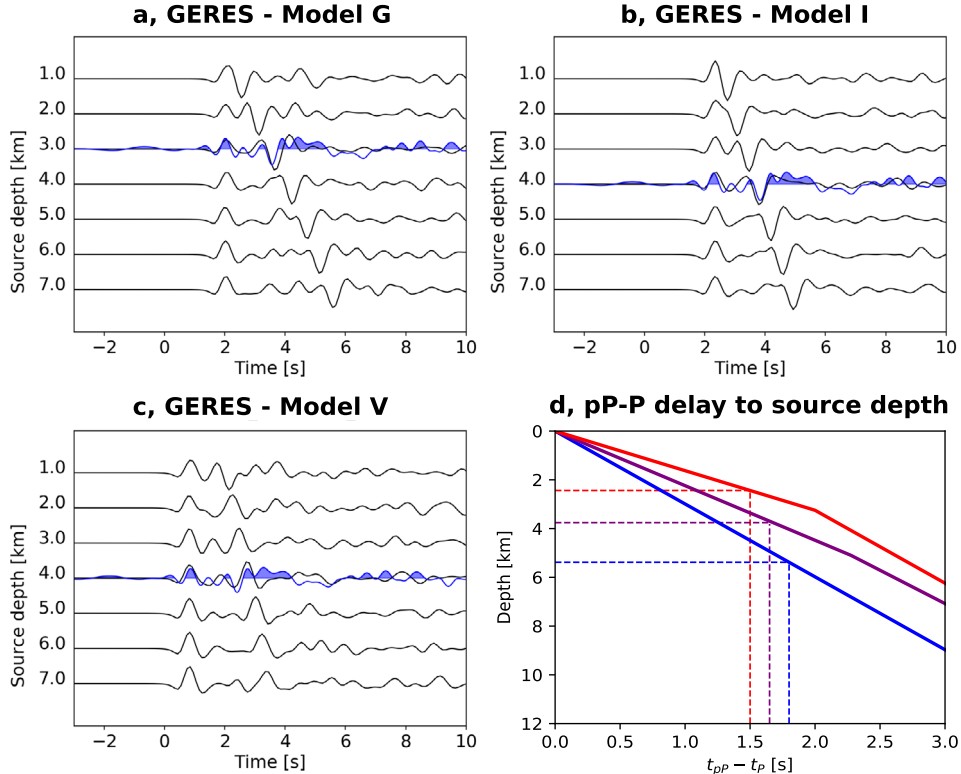

**Fig. 4 Summary of seismic depth estimations for the October 1, 2013, 03:32:44 UTC, earthquake based on seismic array beam modeling.**
**a–c** Comparison of the observed (thick blue line) and synthetic (thin black lines) beams at the GERES arrays (Germany). Synthetics are computed for three velocity models (models G, I, and V in **a–c**, respectively), the preferred focal mechanism derived using model G (strike 41°, dip 56°, rake 4°), and depths between 1 and 7 km. Observed beams are plotted at the depths of 3–4 km, for which waveforms and P and pP pulses are best modeled. Both observed and synthetic beams represent normalized displacements, bandpass filtered in the range 0.25–1.80 Hz. **d** Theoretical source depth as a function of the pP delay for the three models at the source (model I in blue, model G in purple, model V in red): given the pP-P delay, and assumed a velocity model, we can estimate the depth. We consider delays of 1.5–1.8 s, as picked from beams at GERES, Germany and ILAR, US arrays, and an average of 1.65 s. Three scenarios are shown (dashed lines), leading to a depth of 2.4 (red line), 3.8 (purple line), and 5.4 (blue line) km.

waveforms (Fig. 6). The network-based clustering identifies six compact, spatially separated clusters (Fig. 6 and Supplementary Fig. 16) with waveforms changing gradually along the lineament. A supplementary analysis of P/S and Rayleigh/Love (R/L) amplitude ratios reveals smooth trends along the NE striking lineament (Supplementary Fig. 17), which can be modeled using synthetic seismograms. The results indicate that the location change is the reason for the observed waveform variations among neighboring clusters, not varying focal mechanisms. All of these results imply that the fault geometry remains stable over the course of the sequence, with seismicity occurring in neighboring clusters at different times, supporting the progressive failure of different patches along one common, planar structure. Only clusters at the NE and SW boundaries of the cloud show some differences, which may be attributed to the larger separation of these events, or possibly changes in fault geometry and local heterogeneity at the terminations of the active fault segment.

## Discussion

The current debate about seismological results at Castor principally revolves around the hypocentral locations, their spatial distribution, and the depth range of the seismicity. Absolute locations are indeed poorly resolved, as demonstrated by large changes in location when using different velocity models[28]. Absolute location uncertainties for a single event[28] show a similar pattern, with poor resolution in the NW–SE direction and a trade-off with the hypocentral depth. Such location uncertainties

are attributed to the strongly uneven network geometry, in combination with lateral structural heterogeneities in the Valencia Trough[43]. Multi-events relative location methods, based on differential phase arrival times from phase picks and/or waveform cross-correlations, can improve the location resolution and sharpen seismicity patterns[44–46]. Our results, and specifically those of the cross-correlation-based relocation, resolve how seismicity is distributed along an elongated volume, extending sub-parallel to the coast along a N42°E direction (Fig. 2 and Supplementary Fig. 8). The orientation, shape, and extent of the imaged active region match the reservoir's outline[29,47]. The thickness of the layer affected by seismicity is confined to 2 km only, with good resolution. As earthquakes did not occur above or below this layer, and epicentral uncertainties are well below 1 km, the dip of the fault activated across the layer cannot be resolved from seismicity. Shallow hypocentral depths, within the upper crust, are found in other studies[27,28,32], but their absolute values are inconsistent. The delay times between direct and surface reflected sub-vertical waves is only 1.5–1.8 s. We model both the delay and the waveforms (model G) of pP versus P phases for the three largest earthquakes of the Castor sequence using independent data and can constrain the absolute depths at 3–4 km. The centroid depths from our probabilistic moment tensor inversions are also consistent between 0 and 4.7 km (Supplementary Table 2). Deeper sources at 6–8 km have been recently suggested, based on the modeling of short period crustal reverberation in a shallow, low velocity layer[32]. Such an approach also has the potential for accurate depth estimate; however, depth

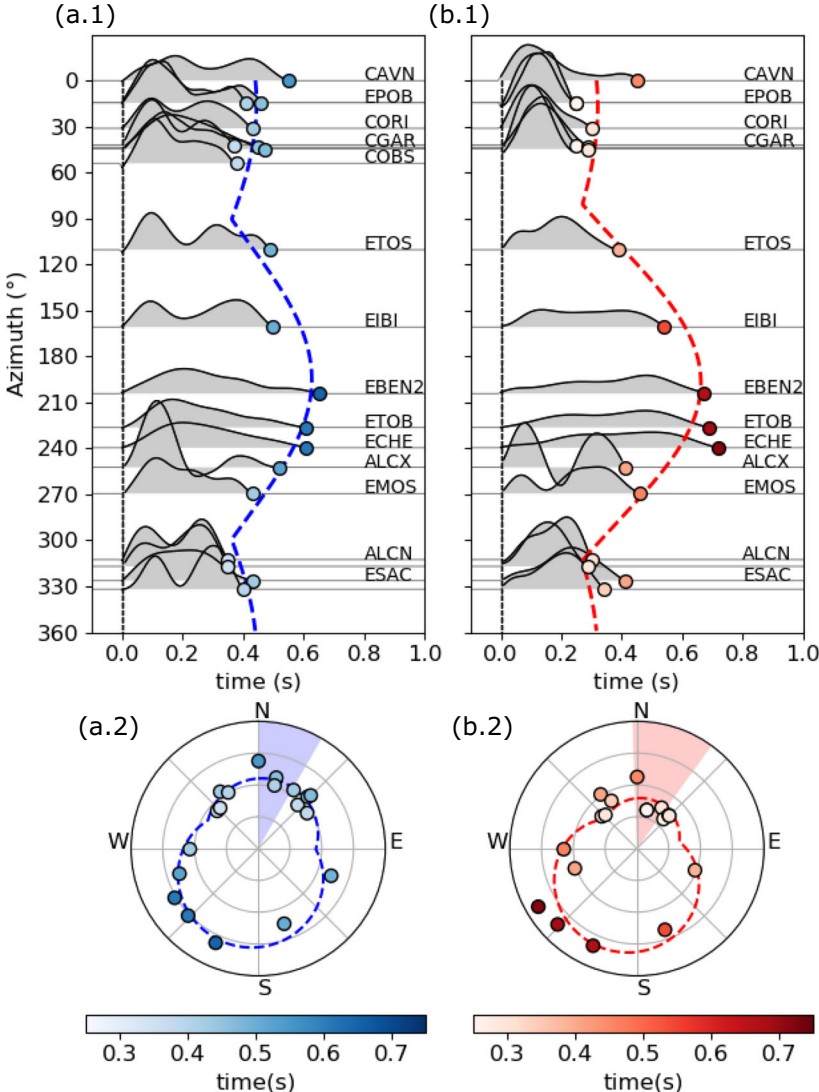

**Fig. 5 Rupture directivity inferred for the two largest earthquakes of the sequence. a** Mw 4.1 October 1, 2013, 03:32 UTC and **b** Mw 4.1 October 2, 2013, 23:06 UTC earthquakes at the Castor platform. a.1, b.1. The duration of apparent source time functions are plotted sorted by azimuth, showing apparent durations (circles) and moment rates (gray area) at each station (see labels), and the asymmetric bilateral rupture model from the directivity analysis (dashed lines). a.2, b.2. Apparent durations as in a.1, b.1 (circles) along with the synthetic predictions for the inverted model (dashed lines) show the predominant rupture direction in a polar plot, where apparent source durations are plotted along the radial axis (axis ticks correspond to 0.2 s). The uncertainty in the direction of the rupture direction (range of angles denoted with blue and red areas) are estimated from the residuals.

uncertainties were not reported there, and it is unclear to which extent this type of modeling is affected by the adoption of a simplified 1D model. Hypocenters at such depths would imply a delay of the pP phase of 2.6–3.3 s, when assuming the local model G, which are too large compared to the delays of 1.5–1.8 s observed for the largest events from regional and teleseismic array analysis. Thus, we conclude that earthquakes were shallower, most likely in the range 3.0–4.0 km but not exceeding 5.3 km (Fig. 4d). Our new Bayesian depth estimates imply that earthquake foci would be only ~1–2 km deeper than the injection point at ~2 km depth at the top of the gas storage layer. As for the moment tensors, our results agree with previous findings[27,31], which are characterized by predominantly strike-slip mechanisms, with fault planes oriented N42°E and N47°W, respectively. We provide explicit uncertainty estimates for all resolved source parameters, which account for both data- and model-related uncertainties. This allows a robust joint interpretation of the results from different techniques, taking into account the resolution of each method.

Our results have important implications for the identification of the activated fault(s). Previously, it has been hypothesized that the seismicity at Castor might have occurred along mapped faults, such as the large N25°E-oriented[48,49] Amposta fault[29] and the Montsia fault system at the reservoir site, which has multiple sub-parallel faults oriented NW[27]. Alternatively, unknown faults with the two possible orientations identified by the fault planes of the strike-slip focal mechanisms have been invoked: either a NE striking fault, gently steeping toward SE[27], or NW trending fault(s)[28]. Our results, both considering the spatial elongation of relative locations, similarity of moment tensors, the high correlation of waveforms and waveform attributes along the trend, and the rupture directivity for the largest events, suggest the activation of a single NE–SW-oriented fault. Our results also exclude the activation of the shallow Montsia fault system or an unknown deeper fault with similar NW–SE orientation[28], below the Amposta fault. Simultaneous activation of the whole Montsia system, which could potentially explain the spatial distribution of seismicity[27], can be rejected as incompatible with the rupture

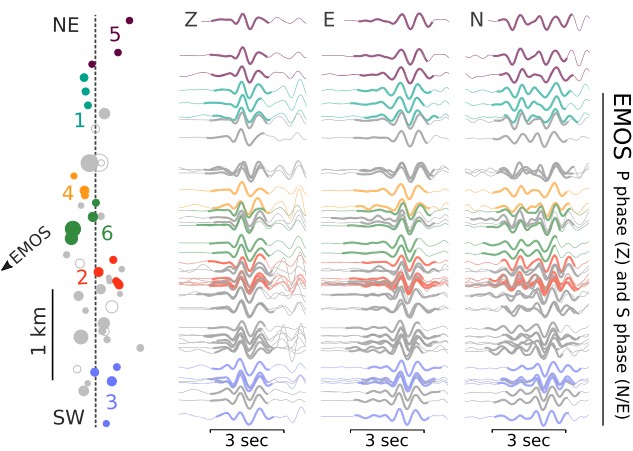

**Fig. 6 Waveform similarity of relocated events.** P and S waveforms (here shown for station EMOS) are very similar, even when filtered with a passband of 1–8 Hz. The signals change gradually along the SW–NE trending lineament (equivalent to profile CD in Fig. 2a). By applying a waveform-based event similarity clustering algorithm, we find six spatially separated clusters of events (color-coded, cluster numbering based on origin time of the first clustered event); note that the clustering is not only based on the waveforms of the exemplary station EMOS, shown here, but also on several other stations. Waveforms are scaled by the maximum amplitude within the time window (bold) and aligned by the cross-correlation time shift to emphasize waveform similarity.

directivity result. The spatial distribution of epicentral location resembles quite well the orientation of the Amposta fault, which forms the roof for the reservoir, and, as so, to the reservoir geometry. However, the Amposta fault dips to the NW, with a ~60° dip angle at shallow depths[48,49], ~40° at the reservoir level, and ~30° below it[29], which is inconsistent with previously proposed focal mechanisms[27,29,31], as well as with all our best quality (A and B) moment tensor solutions. We conclude that the Amposta fault did not participate in the seismic sequence, confirming previous results[27,28].

Only one rupture scenario remains consistent with all results: activation of a previously undetected fault antithetic to the Amposta fault, with a similar strike, but dipping toward the SE. This hypothesis has been previously proposed[27] and can now be refined. The lateral distribution of the seismicity, which resembles the trend of the Amposta fault, and the shallow depth of the hypocenters, just 1–2 km below the injection, points to a fault within or rather below the reservoir, dipping SE, and bounded to the NW by the Amposta fault; such fault orientation and mechanism type are consistent with the local stress orientation[27]. The spatial association of the hypocenters and injection points and the clear temporal correlation among injection operations and seismicity make a clear case of induced seismicity[50], which indeed has never been debated.

Joining the information of relative locations and template matching, we can reconstruct the spatiotemporal evolution of seismicity (Fig. 7). Note that in Fig. 7 we constrain absolute locations in the way that earliest events match the injection location. We can identify three phases. The first begins on September 2, together with the start of the injection, and continues until the injection stops. The second phase begins on September 17 and continues until the end of September and is associated with the evolution of pore pressure and seismicity. These two phases are characterized by different *b*-values, which drops from ~1.0 (larger predominance of small earthquakes) to 0.8 (higher rate of larger magnitude events). The third phase, from the end of

September to early October, includes all of the largest events of the sequence. Overall, waveforms of earthquakes occurring in different phases are very similar (Fig. 6); minor waveform changes can be attributed to slightly changing locations.

During both phases 1 and 2, seismicity migrates unilaterally toward the SW (Fig. 7c), at an average velocity of ~180 m/day, consistent with a unilateral pressure diffusion model[51]. In the hypothesis that the diffusion is confined to a linear structure, the temporal migration of the seismicity front can be reconstructed (Fig. 7b) assuming a diffusivity of approximately 0.5 m²/s[51]. It is noteworthy that the gas is injected close to the platform location and to the SW of it through 4 wells spanning about 600 m, where the reservoir roof is shallowest[29,47]. The horizontal distribution of epicenters resembles the reservoir shape[29,47]. The temporal migration of the seismicity can be explained as follows (Fig. 8). The gas was injected into the highest area of a pent roof-shaped reservoir, elongated NNE–SSW, bounded on the NW by the Amposta Fault and becoming progressively deeper toward SSW and NNE[47]. It has been interpreted that the northern end of the Amposta fault terminates at ~2 km depth into Tertiary layers, suggesting that this portion of the fault is tectonically inactive[29]. Due to the slight inclination of the ridge region of the sealed reservoir in the SW and NE direction, the gas saturated volume would slowly extend to depth during phase 1, to occupy a narrow volume elongated NNE–SSW[29,47]. As the injection proceeded, moderate magnitude seismicity gradually migrated to the SW of the injection point but not toward the NE (Fig. 7). The lack of seismicity to the NE suggests differences in the permeability or in fault(s) properties, such as a fault bending or offset. The seismicity migration continued during phase 2, following the end of injection, probably reflecting ongoing pore pressure redistribution.

The third phase began abruptly by September 28–29, 2013, characterized by a fast (~1 km/day) backward propagation of the seismicity from the extreme SW end toward the injection point, accompanied by all of the largest events of the sequence, including the three M 4+ earthquakes (Fig. 7). Phase 3 does not mark a substantial change in the *b*-value (0.8). In detail, the initial activity appeared at a few spots located on the edges of or between the rupture areas of the later, larger events (Fig. 8). Estimated rupture areas of large events, except the two largest ones, are generally non-overlapping. The assumptions here are a constant stress drop (3 MPa), which leads to rupture sizes consistent with lengths up to 1.0–1.2 km inferred from ASTFs, and that they are on a single plane. However, fast migration and similar waveforms and focal mechanisms for earthquakes suggest a single fault failing progressively, with homogeneous geometry, pointing to a SE dipping, shallow (~3 km) fault plane. The NE unilateral directivity resolved for the largest earthquakes in early October supports the spatial evolution of the rupture process from SW to NE. The last events of the sequence, located at the NE tip of the activated fault, are most likely aftershocks of the largest early October events and are located in the forward directivity direction. Overall, the seismicity evolution marks a kind of round-trip, where a slow diffusion-driven migration away from the injection wells is followed by a sudden return by the seismic triggering of unbroken asperities.

Based on the space–time patterns, several scenarios are possible to explain the complex rupture migration. One is simply pore pressure diffusion, as the earliest activity occurs at the North, where the platform is located, and close to the injection well bottom. Seismicity migrates to the SW at a velocity of about 180 m/day. This is a typical range for pore pressure diffusion[51] and may represent the movement of the pressure front, causing critically stressed areas to rupture first, as the pressure increased. After the injection stops, the SW migration continues as pore

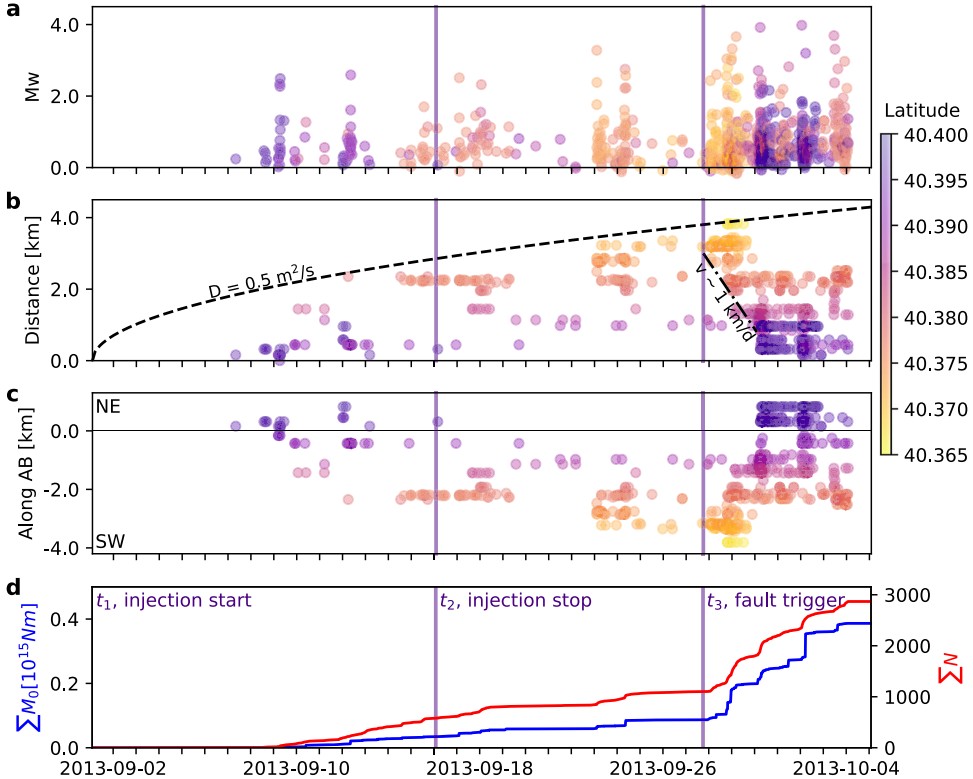

**Fig. 7 Timeline of injection and seismicity.** Temporal evolution of **a** magnitudes, **b** epicentral distances from the injection well, **c** epicentral location projections along profile AB, as in Fig. 2, and **d** cumulative moment release and cumulative number of earthquakes for the time period from 2013-09-02 00:00 until 2013-10-06 00:00 UTC. The onset of three main phases are marked by indigo vertical bars, denoting injection start ($t_1$, 2013-09-02), injection stop ($t_2$, 2013-09-17), and approximate first triggering of the fault hosting largest earthquakes ($t_3$, 2013-09-28, 16:00). The color scale used in **a–c** refers to event latitudes (top right). A dashed line denotes the diffusion curve from the injection point at time $t_1$, assuming a diffusivity of 0.5 m²/s, which explains the migration of seismicity during phases 1–2, and a dashed-dotted line the faster earthquake triggering backward propagation with a velocity of $v \sim 1$ km/day during phase 3. Note that plotted seismicity corresponds to 964 events: 51 accurately relocated events (Fig. 2a) and those detected by template matching using the 51 events as templates; we attributed to detected events the same locations as their templates.

pressure gradients are still present to further drive the pore pressure front in the reservoir. The reverse pattern in phase 3 is different. The occurrence of the largest events progressively rupturing previously unbroken patches from SW to NE. The migration velocity is about five times faster, in the range of 1 km/day. Our analysis indicates that large asperities successively rupture in large magnitude earthquakes. The asperities may have been loaded to more critical stress during phases 1 and 2, where stress in between neighboring asperities was transferred. A possible cause of the rapid backward migration of the earthquakes could be the gradient of vertical stress developed by the continuous exchange of water by gas in the uppermost part of the reservoir. Because of the inclination of the cap rock, which dips gently to the SW, the vertical gas column at the injection point is higher than at the southwestern end of the injected gas layer. Normal stress is smaller at the injection point than at the southwestern tip. A rupture progression toward higher Coulomb stress change has been suggested as the inducing mechanism of the seismicity[52]. The sequential rupturing of asperities could be additionally favored by a progressively reduced fault friction, as water displaced by the injected gas would flow away into fractures and permeable paths, such as the activated fault. Fault valving, involving pressure changes and unsteady fluid migration along fractured regions and faults[53], can also partially explain the alternation of seismicity bursts and quiescence periods. This could also imply the occurrence of aseismic slip, for which we have no direct evidence so far. The clustering analysis shows that weak earthquakes in phases 1 and 2 have the same mechanism as

the larger ones in phase 3 (Fig. 6), and occurred along the same SE dipping structure. If aseismic slip on the Amposta fault accompanied the initial phases of seismicity, as has been recently suggested[34], it did so completely aseismically.

Induced seismicity is a complex issue of public interest, where the exploitation of natural resources needs to be performed in a sustainable way, while also carefully assessing the risk. Within the research field of induced seismicity, scientific outcomes are directly relevant for regulators, industry, and society. In this framework, scientific results need to be explained with a rigorous, non-technical and broadly understandable language[6]. A scientific discussion is desirable and intrinsic to science, but the communication of incomplete and contradictory results may impede explanation of scientific findings to a broad community. A particular problem is a lack of communication about uncertainties, both epistemic and aleatory, which limit our scientific answers to societal questions. This was probably the case for the Castor project, where different scientific publications claimed apparently robust answers to questions that remained debated.

Besides providing new analysis and results on the processes controlling the Castor seismic sequence, the present work aims to harmonize previous results and their apparent discrepancies. Using advanced, waveform-based seismological methods and combining local and regional seismic data, plus seismic arrays at larger distances, allows us to reconstruct the rupture process of moderate earthquakes offshore and thus to shed light on the Castor seismic sequence, proving that this is possible despite the lack of a dense local network. Seismic monitoring regulations

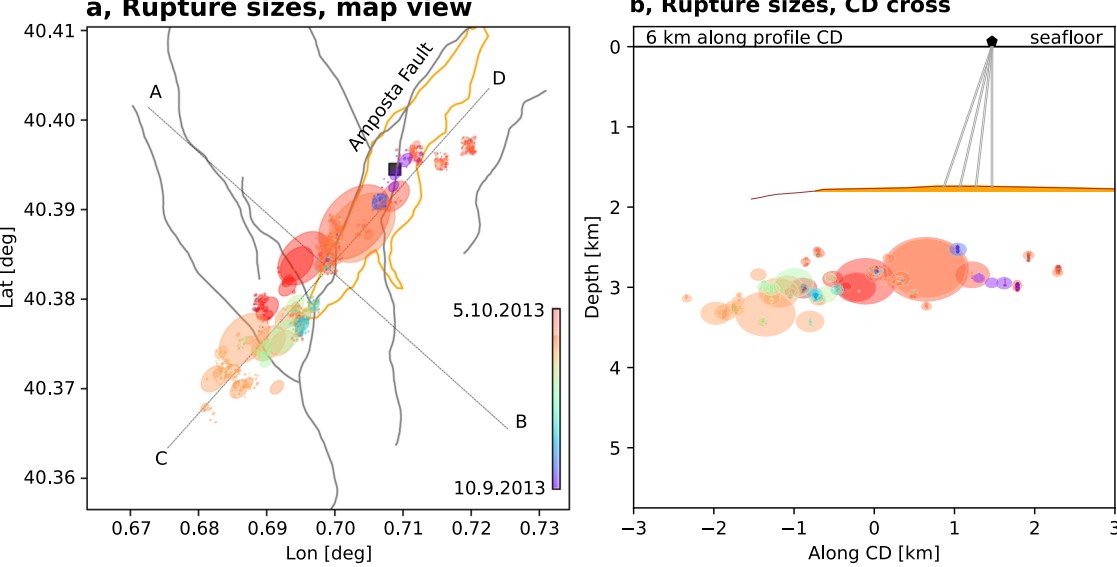

**Fig. 8 Temporal evolution of earthquake rupture size.** Rupture sizes are shown in map view (**a**) and along CD cross-section (**b**). During the SW migration in phases 1 and 2, which accompanies the spatial extension of the injected gas volume within the NE–SW oriented reservoir, the seismicity rate is discontinuous and small-size earthquakes alternate with short periods of quiescence, leaving large unbroken patches. These are filled by larger earthquakes occurring during phase 3, when seismicity migrates backwards. We assume circular rupture areas, with sizes plotted according to a crack model for a fixed stress drop of 3 MPa, and color denoting origin time (see color bar in **a**). Seismicity and locations are from the catalog based on join relocation and template matching, as in Fig. 7. Small, random location shifts have been applied for events below magnitude 2.5, mostly detected by template matching, to better visualize their overlapping locations. The orange line in **a** marks the approximate outline of the reservoir at the depth of 1800 m[29] and thus the rough extension of the gas-filled volume. A brown line and an orange filled region in **b** sketch the projection of the reservoir roof and gas-filled volume along the CD profile, respectively. The Castor platform is marked by a black square in **a** and a pentagon in **b** (here gray double lines sketch the path of injection wells)[29].

recently adopted in different countries[54,55], ensuring adequate induced seismicity monitoring at a local scale, can support a prompt and accurate analysis of future cases of induced seismicity, a deeper understanding of their seismogenic processes, and a clearer and harmonized communication of scientific results to the society.

## Methods

**Seismic catalogs, seismic data, and velocity models.** The seismic catalog by the National Geographic Institute of Spain (IGN) includes 536 events in the time period 5.9.2013–15.10.2013 in the study area (http://www.ign.es/web/ign/portal/sis-catalogo-terremotos, last visited 1.10.2020). The catalog compiled by the Ebre Observatory, using also stations ALCN and ALCX, includes 938 events. Both catalogs were used in previous publications[27,28] but present some inconsistencies in terms of magnitudes and locations. For example, the IGN catalog counts 148 events with MbLg >2.0 and 5 with MbLg >3.0, while the Ebro catalog has 67 events with Ml >2.0 and 13 with Ml >3.0. A drop of $b$-values from $1.4 \pm 0.2$ (injection phase) to $0.9 \pm 0.1$ (post-injection phase) has been identified using the Ebro catalog. A similar reduction is also found using the IGN catalog ($b$-values of 1.5 and 1.2), after computing the completeness magnitude using the maximum curvature technique[56]. Differences among absolute $b$-values using the two catalogs are attributed to the mentioned magnitude discrepancy. Seismic data (Supplementary Fig. 1) and velocity models (Supplementary Fig. 2) used in this work are listed in the Supplementary Notes 2 and 3.

**Earthquake detection.** Waveform matching is nowadays widely applied to detect earthquakes and other seismic sources, with the capability to find weak signals below the noise level. We adopt here the template matching software PyMPA (https://github.com/avuan/PyMPA37, last visited 1.10.2020), able to run the cross-correlations in overlapping moving windows on years of continuous waveforms and using thousands of high-frequency templates[57]. PyMPA was used in previous applications, such as for the processing of recent seismic sequences in Italy[58–60]. Here we apply it to ~1 month of continuous data, from September 9 to October 4, 2013. We use 148 earthquakes (all those with M≥2.0) as templates and process seismic data of EBR and ICGS networks in the frequency range 2–8 Hz. Three-component continuous waveforms from stations ALCN, ALCX, CMAS, and COBS are downsampled to 25 Hz. Templates are trimmed using a 10 s data window, starting 3.5 s before the theoretical S wave arrival. Travel times are computed using the ObsPy port[61] of the Java TauP Toolkit routines[62], assuming the crustal model G[28]. We adopt Kurtosis-based tests to evaluate the signal-to-noise ratio (SNR) of

templates[63] avoiding the use of unwanted signals in the matching technique[57]. Earthquake magnitudes for the extended catalog are computed relative to the magnitude of large master events, given in the original Ebro catalog, considering that the scalar moments of two events with similar waveforms scale as the amplitudes of their waveforms.

**Relocation based on waveform cross-correlation.** Relocations are first obtained using a linear master event location approach, assuming parallel rays from all events to a given station (Fraunhofer approximation), which is appropriate for clustered seismicity. As a master event, we use the earthquake on October 4, 2013, 09:55 UTC. This event shows optimal SNR at all stations, is among the largest earthquakes in the sequence, but still does not show any signature of source finiteness. Indeed, its waveforms look simple, and there is no evidence for the superposition of further, smaller events in the waveforms. From a visual inspection, its waveforms represent something close to the average waveforms of the sequence. Consistently, we will find a posteriori that its location is near the center of the sequence. Data of 51 largest earthquakes were manually re-picked, and S phase identification improved after comparing waveforms for different events, applying a 2 pass 4 pole Butterworth bandpass filter in the frequency band 0.7–2.0 Hz. This simplifies the waveforms considerably, while preserving a reasonable SNR; in this frequency band, P and S phases from all events look generally similar for each station and component, with the only exception of S waves at ALCX; S phases can be picked well at ALCX, but waveforms appear sensitive to minor location changes, which we attribute to the fact that S phases are nodal at ALCX, given that the WSW–ENE orientation of the tension axis for the typical focal mechanism at Castor (Fig. 3) is similar to the azimuth of station ALCX. For each station, we selected the best components showing the most consistent and clear P (Z component in most cases) and S (one horizontal component, except for station ERTA) pulses, which are finally used to compute cross-correlations and derive differential times. For the cross-correlation, we use 4 s long time windows centered at the P and S arrivals. We do not make use of station COBS for the relocation, after verifying a general low correlation of P-waves, which is probably due to noisy signals at the OBS. The correlation is done twice: in the first iteration of the relocation, windows are cut using as reference the estimated arrival times and larger lags up to ±1 s are allowed, while in the second iteration they are cut using as reference the delay time estimated after the first run, and the correlation is only allowed for fine tuning. Waveforms are then aligned according to the cross-correlation delay times and a final visual inspection is done: noisy traces and all waveform showing evident misalignments (e.g., focusing on different phases) are excluded. All remaining waveforms contribute to the relative location. In our work, we discuss the relative locations for 51 earthquakes, requiring that a minimum of 5 relative time readings are available. Among the stability tests to verify the

relocation approach, a jackknife analysis showed that the overall appearance of the sequence does not change when removing one station.

**Relocation using S-P differential times and distance geometry techniques.** The monitoring network at Castor has a large azimuthal gap and the closest station is located onshore at ~20 km distance from the injection wells and the seismic cluster; such unfavorable geometry challenges accurate hypocentral locations using conventional standard location techniques. To overcome these limitations, we apply a new location method, designed to locate clustered seismicity using only one or a few seismic stations[36]. The method is based on the solution of a Distance Geometry Problem, which consists in finding the coordinates of a set of points by using the distances between some point pairs. This approach is particularly suited for the analysis of clustered seismicity, occurring outside of a seismic network[36]. Input data are the inter-event distances between earthquakes pairs, which are estimated using the differential S-P arrival times at two seismic stations nearly perpendicular with respect to the centroid of the cluster[36]. To ensure the solution uniqueness and bind the locations to an absolute reference frame, the method requires at least four non-coplanar seismic events, for which the absolute hypocentral coordinates have to be known in advance; the location of the remaining events is then performed iteratively, one event per iteration. At Castor, we relocated 408 earthquakes (Supplementary Dataset 3) by using only the two closest seismic stations, ALCN and ALCX. These stations offer a very good set-up for this method, as they are located in an almost perpendicular direction with respect to the seismicity cluster[36]. Poor location of the reference (master) events may have pernicious effects on the location results[36]. For this reason, instead of using the absolute locations of four non-coplanar master events, we only rely on the absolute location of one master event, which is used as origin of the reference system, and the relative locations of three more master events; their relative location can be estimated using their inter-event distances. By solving the seismological distance geometry problem[36] with this semi-relative reference frame, we can first reconstruct the internal shape of the seismicity cluster but not its orientation. Finally, to find the orientation of the seismicity cluster we perform a grid search over all possible cluster orientations (i.e., performing rigid rotations of the cluster for different azimuthal angles), minimizing the differences between theoretical (model G averaged) and observed differential S-P times for each earthquake within the cluster at the stations ALCN and ALCX using an L2 norm and maximizing the rectilinearity of the $t_S$-$t_P$ versus source-station distance plot by using the Principal Component Analysis. This relocation process has been performed with the software HADES[36] (https://github.com/wulwife/HADES, last visited 01.05.2021).

**Centroid moment tensor inversion.** Moment tensor inversion for the Castor sequence has been performed in previous studies[27,29,31], mostly by fitting regional low-frequency full waveforms in the time or frequency domain or derived by simple first motion polarity analysis; source parameter uncertainties were rarely reported. Here we improve previous results by performing a deviatoric moment tensor inversion for the 11 largest events using Grond[37,64], a probabilistic earthquake source inversion framework. The following precautions were taken to provide accurate and robust solutions: (a) we rely on a broader range of seismic observations, from a larger number of seismic stations (up to 28 stations against a maximum of 7–17 in previous studies), (b) we simultaneously model stations located inland Iberia, on Balearic islands as well as at one ocean bottom station, using different velocity models and fitting procedures to account for significant differences among ray paths and station installations, (c) we test the effect of different crustal velocity models, and (d) we discuss the uncertainties of each source parameters quantitatively. Data preprocessing included seismic data deconvolution by the seismometer transfer function, integration to displacement, demean, and detrend. Data quality was manually assessed, and a few seismic traces were removed, in presence of gaps or poor SNR. We fit 3 components (vertical, radial, transversal) seismic data, filtered in the frequency band 0.04–0.10 Hz. We fit simultaneously the following observations, equally weighted, using an L1 norm: (1) full waveforms cross-correlation for stations up to 350 km distance, (2) full waveforms in the time domain at closest stations, below 100 km, and (3) full waveform amplitude spectra in the frequency domain up to 350 km. Seismic stations at Balearic islands are only used for the frequency domain amplitude spectra fit, because the velocity models do not reproduce the crustal structure along these ray paths and thus synthetics cannot reproduce the complexity of the observed seismograms in the time domain[27]; amplitude spectra inversion has proven to constrain moment tensor solutions for offshore locations and to be less affected than time domain inversion to approximated velocity models and asymmetric station geometries[65]. Furthermore, for the seafloor seismic station COBS we only fit the cross-correlation, because of the unknown station coupling at the seafloor, which affects the observed seismogram amplitudes. While not contributing to the magnitude estimation, fitting cross-correlation helps to resolve the moment tensor geometry[64]. Using COBS data helps reducing the seismic gap eastwards. Grond performs a bootstrap over the seismic data, and resulting solution ensembles are used to estimate parameter uncertainties; we report both the best solution, obtained fitting all data, and a mean solution upon 100 bootstrap chains.

**Array-based source depth.** The determination of the hypocentral depth for offshore earthquakes is difficult, when most of the seismic stations are located inland, with poor azimuthal coverage. In such conditions, a trade-off may exist among origin time, distance from the coastline, and hypocentral depth, and location uncertainties are often characterized by very elongated ellipsoidal confidence regions, with a poor resolution both along the direction perpendicular to the coast and as a function of depth. This pattern is well seen, for example, for single earthquake location uncertainties at Castor[28]. Furthermore, a robust and precise depth estimation requires accurate knowledge of the crustal structure, which is not fully agreed at the moment, with variable models proposed, including 1D models listed in this paper, and a 3D model[28]. Here we use a different approach, specifically focused on the determination of the hypocentral depth. We compare observed beams recorded by seismic arrays at teleseismic locations with synthetic beams for different depths. In particular, we aim at modeling first P onsets and later pP and sP phases reflected at the seafloor. The delay between the first P onset and surface (sea bottom) reflected pP phase is given by the time needed by a P phase to travel two times across the shallow layer above the hypocenter. Consequently, this delay increases monotonically as a function of source depth and its exact relation to source depth depends on the crustal profile of P velocity; an example considering crustal models used in this study is shown in Fig. 4d. This approach has several advantages for the study case. First, assuming that the seismicity is located in the vicinity of the Castor platform, as it is currently commonly agreed, we only need to model the crustal structure at this site, which is pretty well known based on the preliminary work of the Castor project, and we can also ignore the complex, 3D structure of the Gulf of Valencia. However, the synthetic beams consider the crustal structure at the array different from the structure at the source. Next, the largest events of the sequence, with a magnitude Mw 4.0–4.1 are energetic enough to be well recorded at seismic arrays located at thousands of km distance, while weak enough that their short rupture durations (<0.5 s according to Empirical Green's function results discussed in the following) allow separating well P and pP onsets. Observed beams are qualitatively compared to synthetic ones, which can be computed for the desired moment tensor, specific source, and receiver models (a global mantle model is used for the wave propagation between source and receiver regions) and for a range of hypocentral depths. The method has been previously successfully applied to other shallow seismicity cases[5,66,67], resolving the hypocentral depth in some cases by <1 km.

**Rupture directivity.** The rupture process signature for the largest earthquakes can be retrieved by applying EGF techniques[39]. Seismic waveforms of selected seismic events in the sequence, i.e., those sharing similar location and focal mechanisms as the target events and having a weaker magnitude (~1 order magnitude lower), can be used as EGFs, in order to accurately model the propagation between the focal region and the seismic stations. ASTFs are retrieved independently at each station by signal deconvolution[39,52,68], whenever both EGF and mainshock are recorded with high SNR. The ASTF reveals the apparent duration and its azimuthal variation can be used to detect a predominant rupture direction, so-called rupture directivity. The joint interpretation of focal mechanism and rupture directivity can help to solve the fault plane ambiguity and to identify the rupture plane orientation. We select the two largest earthquakes of the Castor sequence as targets (Supplementary Table 3) and five weaker earthquakes as potential EGFs (Supplementary Fig. 15). Manual picking was performed to align waveforms of the target earthquakes and corresponding EGFs; picking errors are estimated <0.05 s. We perform the signal deconvolution in the frequency domain by spectral division. To avoid numerical instabilities derived from the deconvolution, a Gaussian lowpass with pulse width of ~0.25 s and a water level of 0.01 of the maximum spectral amplitude are used[68]. ASTFs are normalized to unit area according to the total seismic moment of the target earthquakes, and negative values derived from the deconvolution are removed. We prove the ASTF stability by testing different EGFs (Supplementary Fig. 15): a single pulse can be identified for both target events, with slightly shorter apparent durations at NW and NE azimuths. We finally select the October 4, 2013, 09:55 Mw 3.5 earthquake, which shows good SNR at all stations, as the best EGF candidate and use it for S wave windows (window lengths 10–15 s). We have a good azimuthal coverage using regional stations up to 275 km distance (Supplementary Fig. 1). Finally, source parameters associated with a line source, such as rupture length, total duration, and percentage and direction of slip asymmetry, can be retrieved[69]. Parameter uncertainties, e.g., with respect to the rupture direction (Fig. 5), are estimated from the residuals[68]. Theoretical predictions for unilateral and asymmetric bilateral rupture are considered to adjust the azimuthal pattern of the apparent durations identified from ASTFs[52]. We test different rise times and choose values of 0.15 and 0.25 s, which lead to rupture velocity estimates of ~2.7 km/s (Supplementary Table 3).

**Waveform-based event similarity clustering and amplitude ratio analysis.** The event clustering is performed using the waveform clustering toolbox CLUSTY[42]. First, event network similarities based on waveform cross-correlations from three-component traces at selected stations (ALCN, ALCX, EMOS, EPOB, CMAS) are computed. Raw waveforms are filtered with a passband of 1–8 Hz to resolve small variations in the P and S waveforms. We use 1.25 s long time windows, starting before the picked phase arrival. The network similarity matrix is used as input to a density-based clustering, which identifies events with common P and S waveform characteristics. We tuned the clustering parameters using different provided metrics, e.g., the silhouette score[42] to ensure that the clusters are well separated (Supplementary Fig. 16). In the subsequent amplitude ratio analysis (P/S and R/L,

respectively, Supplementary Fig. 17), we automatically determine the maximum amplitudes in the given windows for P and S phases. For surface waves, the maximum amplitude of the Rayleigh wave, within the time window defined by group velocities of 3.5 and 1.5 km/s is extracted from the vertical component. The Love wave amplitude is read from the transversal component after rotation of the horizontal traces. For the modeling of R/L ratio trends, we calculate synthetic seismograms for each event along the lineament at stations EMOS and EPOB and repeat the procedure described above.

## Data availability

Seismic data used in this study are available by open web services (https://geofon.gfz-potsdam.de/waveform/webservices/, https://www.orfeus-eu.org/data/eida/webservices/, https://www.icgc.cat/en/Downloads/GeoData/EIDA-webservices) or hosted at IGN and Ebro data centers, where access can be obtained upon request; till today, only data of stations ALCN and ALCX are protected and are not available due to data privacy laws; access can be obtained, upon a justified, written request to the data owner, the Ebro Observatory. The advanced data generated in this study (i.e., different seismic catalogs including resolved source location and source parameters) are provided as Supplementary Datasets 1–3.

## Code availability

Analyses are performed using established codes and providing all details to reproduce our analysis. Codes used are in most cases open source (e.g., template matching, depth phase modeling, moment tensor inversion, waveform-based clustering); in all cases, specific references are provided, including methodological details.

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

## Acknowledgements

We are thankful to Dr. J. V. Cantavella and IGN for providing waveform data and metadata of the network ES. S.C. and P.N. received funding by the European Union RFCS project PostMinQuake grant 899192. J.A.L.-C. was financed by the European Union's Horizon 2020 research and innovation program under the Marie Skłodowska-Curie grant agreement 754446 and UGR Research and Knowledge Transfer Found–Athenea3i and the Deutsche Forschungsgemeinschaft (DFG, German Research Foundation) - Projektnummer (407141557). D.S. received funding by the Spanish National FEDER/MINECO Project PID2019-109608GB-I00/AEI/10.13039/501100011033, FEDER/Junta de Andalucía project A-RNM-421-UGR18 and Research group RNM104 of the Junta de Andalucía. F.G. was financed by the European Union's Horizon 2020 Framework Programme under the Marie Skłodowska Curie Grant agreement (790900).

## Author contributions

S.C., D.S., T.D., and W.L.E. contributed to the work conceptualization. A.V. contributed to the template matching and catalog analysis. D.S. contributed to the relocation based on cross-correlation. F.G. contributed to the relocation based on differential times. S.C. contributed to the depth estimation based on differential times and the moment tensor inversion. J.A.L.-C. contributed to the directivity analyses. P.N. contributed to the waveform-based clustering. E.B. contributed to data processing and catalog preparation. S.C. and J.A.L.-C. elaborated the graphics. S.C. contributed to the original draft, reviewed and edited by all coauthors.

## Funding

## Competing interests

The authors declare no competing interests.
