## [Peer Review File · Nature Communications]

REVIEWER COMMENTS

Reviewer #1 (Remarks to the Author):

In general, I think the paper is very interesting, gathering a lot of cutting-edge methods to get most out of the data and to better characterise and understand the development of induced seismicity at Castor. The detailed analyses that were carried out could be applied likewise to other dataset in the future, so the work deserves to be published.

Comments and questions below:

* Introduction: l.67 "not exceeded"  did not exceed.

Isn't the Groningen gas field fulfilling the criteria here? (M3.5 in 2006, M3.6 in 2012, M3.4 in 2019)?

l.102 "more clear"  clearer

* Results

1- Template matching: l.138 "with the aim of better trackING"

How were the 148 template events chosen? Magnitude distribution of the templates?

Could Fig.S3 be improved? Since it is in supplementary material, there is no space limitation, and currently it is quite hard to see the waveforms properly. Are the red traces the template waveforms? What are the dashed vertical lines? What does "preferred channels" mean in the caption?

2- Earthquake relocations

In the first sentence (l.152), could you already name the 2 methods that you are going to use here? Makes it easier to follow afterwards.

You write that the methods are well-suited for clustered seismicity, but it hasn't been introduced yet at this stage of the manuscript. Maybe it would make more sense to have the section about clustering earlier, just after the template matching? Also, I might have missed something here, but why are there only 51 events relocated with the first method? Since you used template matching to detect many more events, there might be a lot of well-correlated events. Why aren't they used further? Is the quality not good enough (low SNR)? Especially, you relocated 51 events but used 148 templates for the detection...

Fig.S6: which covariance matrix?

Later on, in the method section, l.451, you also refer to fig.S6, but are the results from the jackknife analysis visible there?

About the second approach: did you relocate the 51 events as well? If yes, how do the locations compare?

More generally, the comparison is difficult to establish in Fig.2, and the seismic clouds look quite different, and particularly in Fig.2b, the 411 events are rather scattered? Wasn't it possible to relocate the same events with the previous method if P- and S-wave picks are available anyway?

3- Moment Tensor Inversion: is there a difference between in Fig.S8 and S10, except that there are more stations? (doesn't need to be changed, I just wonder if I missed something)

4- Depth estimation: l.187 "Measured time delays (...) ARE"

Fig.S11: l.168 "(other events)" ???  can be removed

5- Rupture directivity: l.203 "the estimated rupture velocity"

Fig.5: I didn't understand what the uncertainty mean (blue and red shaded areas in the polar plots)? Neither how it was obtained, nor how it should be understood?

In the figure caption, l.871, "are plotTED"

5- Waveform clustering: as mentioned above, maybe this section could come earlier?

l.215-216: did not understand the first part of the sentence.

Again the same comment as previously, but couldn't template matching also define some clusters?

Fig.S13a: could you give some CC values here (or plot the CC matrix, or the silhouette plots)? The waveforms are both similar and (I agree) dissimilar in the details. But just out of curiosity, would be great to know how much they vary.

Fig.S13b: the profiles are a bit difficult to situate/understand...

* Discussion:

l.234-235: replace by "such location uncertainties are attributed to (...)"

l.239: "clearly resolve the seismicity": I agree for the 1st relocation method, but less obvious to my eyes for the 2nd method

l.252: change words order  "depth uncertainties were not reported there"

l.275, l.282, l.293: replace "what" by "which" or "that"

l.286: "antithetic": what does it mean in this context? it comes in opposition to what?

l.302: I think there should be a short sentence about the 3rd phase here as well.

l.308: what does "healing backfront" mean here?

Fig.6b: did you use the locations of the 411 events here? Maybe only me, but I don't think the temporal migration of the triggering front is that obvious...

Fig.6d: the scale of the cumulative seismic moment should be adapted! Cannot see anything right now...

Fig.7: the colours are more difficult to distinguish in this figure. May be a better colour map (otherwise it is OK)? Could you label the Amposta fault in this figure again?

l.327: b-value from 0.8 to 0.66. Isn't it a substantial change? Use of 1 vs 2 decimals? Phase 3 also contains many more events than phase 2. How does it affect the b-value that is a statistical parameter+

l.348: "driveS"

l.352: "phaseS"

l.364: "involve a role"  imply

l.374: "the case FOR"

l.381: "proofing"  proving (also l.564)

l.384: "more clear"  clearer

General English correction: I noticed some inconsistencies in the use of past/present tenses.

Reviewer #2 (Remarks to the Author):

Review of "Seismicity at the Castor gas reservoir driven by pore pressure diffusion and asperities loading" by S. Cesca and others

In this paper, the authors re-analyze and supplement data related to a series of $M > 4$ earthquakes at the Castor injection platform. Given the diversity in previously published results surrounding these earthquakes, this more detailed analysis with additional data is not only warranted, but needed, especially since these earthquakes are the results of induced seismicity. For cases of induced seismicity, it is very important for unambiguous results, if possible, that can be communicated to operators, regulators, and the public. As such, I find this an important study. Further, scientifically this is a very complete seismic analysis that warrants publishing--the suite of tools available for studying this type of sequence has been employed in a systematic and rigorous approach. Overall, the paper is well-written. I make a few specific suggestions for clarity below. I also ask for elaboration in the text describing a couple of the methods.

Comments:

The distance geometry (line 459) approach for locating earthquakes is new to me and there are no references. It would be helpful if the authors could elaborate more on this process and perhaps add a figure to the supplement that shows the process. From what I get, which I am not entirely sure is accurate, you use S-P as a metric to find events spatially close and this largely works because the stations used in the analysis are approximately equidistant from the centroid of the sequence. Then starting from at minimum 4 well-located earthquakes you iteratively add one event at a time and relatively relocate all(?) events based on these S-P times.

Questions:

- If the event is not close to the centroid how do you associate the S-P times between the two stations? The distance (S-P time) will not be the same to each station.
- Does the order that you add stations matter? Is there a rule for how close you have to be to one of the 4 well-located events?
- How different are the results in the absolute locations when using the different velocity models?

Adding the array based analysis to get source depth was a great addition! You really needed some independent data to get good depths.

Template matching.

- In Fig S3, there are several stations where the correlations do not look all that good. These do tend to correspond to low-correlation values, but this begs the questions, what is the threshold that you used to declare a detection?
- You also seem to match on the S-wave and from Fig. S3 the P-wave arrival is not all that obvious. Were you able to get P-picks for the relocations?
- Are the blue lines in Fig S3 the theoretical S-wave arrival or the start of the data window?

b-values: In Fig. S5, the fit to the injection catalog is not good for $M > 1.5$. In fact, the fit is only good over 1 magnitude unit, so I do not find this b-value meaningful. My guess is that the M_c value is artificially low perhaps as a result of using magnitudes from the template matching. If you choose to persist with the argument of a b-value difference this section needs additional work and you need to show a fit over the entire magnitude range.

Amplitude spectra (Fig S9): Most of the fits look really good. I was curious about the middle column transverse and radial component, which seem to have a much larger misfit than the other fits. Is this a function of the velocity model or the source properties?

Line 129: it would be helpful to add "(detailed below in Methods section)" after techniques. Given the reference to the supplementary figures, it was not straightforward to me where to find the details regarding the processing, especially with the non-standard format (methods at the end).

Line 172:, please add a reference after inversion approach. It is referenced in the methods section, but it would be helpful to have it referenced in the results section as well.

Fig S4, since there is overlap in the two catalogs, perhaps make the Erbo catalog open symbols, so that extended catalog can be better seen

Reviewer #3 (Remarks to the Author):

General Comments

This a very thorough and detailed manuscript about induced seismicity at the Castor gas reservoir offshore Spain proved to be caused by gas storage injection, which is considered rare. Because of previous work showing mixed conclusions, this study presents some detailed seismological analysis using new approaches that quite frankly would be difficult to improve upon. This manuscript is well written with clear figures, and I have few issues with any part of the manuscript. My main concern is that not enough detail is provided on the techniques used to be able to reproduce the results. The science results are very interesting and important, and overall, this is an excellent paper.

Specific Comments

Some of the detail to the techniques are either not presented, are after the results, and or are in the supplement. I do not know if this is a constraint of the requirements of the manuscript, but I would suggest have the methods sections before results. When I first started the template matching description, and I has no idea how this was done, and then realized after getting through all of re results and discussion, this was described in the methods section at the end of the paper. If this is to remain the structure of the paper, it would be good to add a sentence or to describe in words the approaches.

Lines 152-154: "used in previous papers." What previous papers? Need to cite them here. Again, this is with details to the approach. I have no idea what "advanced relative earthquake localization methods" are being used. (I found it in the methods section... too late for me... confused me to no end).

Lines 171-173. What probabilistic approach was used? Not cited here. How do we know what is "extremely stable"? We have no context if this is standard for the approach. I understand that the choice of models

does not impact result, making it stable. However, the order of the wording is critical here.

Line 184-192. As mentioned in the text, the geometry of the array and the lack of being within a focal depth of the events makes it very difficult to determine depths. I appreciate the effort here to use depth phases. However, I have yet to see an approach for local waveforms that convinces that you are modelling depth phases (perhaps I just have not read the right papers). Green's functions are difficult to calculate at local distances because of local geology complexities. Not having the details of the approach (how synthetic waveforms were calculated) makes this section even less convincing.

Lines 212-227: Figures of this analysis are being presented only in supplemental material. Is it worth presenting if it is only supplemental?

Lines 230-267: Outstanding discussion on depth resolution.

Line 275: "what excludes", should it be "which excludes"?

The methods section should be moved up before results. I find it very difficult to understand the results without knowing what was done exactly (so as to believe the results).

Reviewer #1 (Remarks to the Author):

In general, I think the paper is very interesting, gathering a lot of cutting-edge methods to get most out of the data and to better characterise and understand the development of induced seismicity at Castor. The detailed analyses that were carried out could be applied likewise to other dataset in the future, so the work deserves to be published.

We are thankful for the positive evaluation of our work.

Comments and questions below:

* Introduction: l.67 "not exceeded"  did not exceed.

Isn't the Groningen gas field fulfilling the criteria here? (M3.5 in 2006, M3.6 in 2012, M3.4 in 2019)?

We corrected the text accordingly. Groningen is certainly a fascinating case study, but our aim here was to list only cases of seismicity related to gas storage operations. We reformulated the sentence to make this clearer.

l.102 "more clear"  clearer

Corrected.

* Results

1- Template matching: l.138 "with the aim of better trackING"

Corrected.

How were the 148 template events chosen? Magnitude distribution of the templates?

As templates we used all earthquakes with a magnitude larger than 2. We have now rephrased the text to make this clear.

Could Fig.S3 be improved? Since it is in supplementary material, there is no space limitation, and currently it is quite hard to see the waveforms properly. Are the red traces the template waveforms? What are the dashed vertical lines? What does "preferred channels" mean in the caption?

We split former Fig. S3 into 3 new figures (now Figs. S3-S5) showing only the good quality channels (8 channels) used in the template matching. Generally, COBS station is not useful because the signal to noise ratio is low. We also made more explicit the figure caption. The templates (red traces) are superimposed on the continuous data (black traces) for the declared detection. The dashed blue line represents a theoretical S-wave arrival. It is used only as a reference to trim 10 s template waveforms. Preferred channels are the closest to the event.

2- Earthquake relocations

In the first sentence (l.152), could you already name the 2 methods that you are going to use here? Makes it easier to follow afterwards.

Done

You write that the methods are well-suited for clustered seismicity, but it hasn't been introduced yet at this stage of the manuscript.

We reworded the sentence, here. These two locations techniques are suitable for the relocation of seismicity, when their hypocenters are spatially clustered and not in the vicinity of the network, which is the case of the Castor sequence and also known from previous study. This is different from the waveform clustering performed later, which actually aims at showing that events are clustered based on their waveform similarity.

Maybe it would make more sense to have the section about clustering earlier, just after the template matching?

As mentioned before, the waveform clustering (now renamed as waveform classification of relocated events) recognizes group of events with very similar waveforms recorded at different stations. Such high waveform similarity has implication for the rupture process, as it implies a similar location and a similar mechanism (finally, as we believe, the activation of the same fault).

For this reason, we need to introduce this technique after the discussion of locations and moment tensors.

Also, I might have missed something here, but why are there only 51 events relocated with the first method? Since you used template matching to detect many more events, there might be a lot of well-correlated events. Why aren't they used further? Is the quality not good enough (low SNR)? Especially, you relocated 51 events but used 148 templates for the detection...

The 148 templates, chosen on a magnitude base from the original catalog, are used for template matching, which aims at identifying weak events, which would remain otherwise undetected. As a result of the template matching we have a larger catalog (> 3,000 events). However, not all these events (i.e. not all master events) can be easily relocated (so that the catalog of relocated events includes only 51 events). The reason is that the template matching works with few stations, and can be performed whenever one or few good quality recordings are available, while the relocation procedure requires good quality signals at more stations, which is not always available.

Fig.S6: which covariance matrix?

We simplified the figure caption, removing unnecessary details that are not discussed in the paper.

Later on, in the method section, l.451, you also refer to Fig.S6, but are the results from the jackknife analysis visible there?

We apologize for the inconsistent citation, referring to a working figure which we finally not included, and now removed it.

About the second approach: did you relocate the 51 events as well? If yes, how do the locations compare? More generally, the comparison is difficult to establish in Fig.2, and the seismic clouds look quite different, and particularly in Fig.2b, the 411 events are rather scattered? Wasn't it possible to relocate the same events with the previous method if P- and S-wave picks are available anyway?

We improved the description of the second approach and the comparison between the two locations method, shortly extending the text and improving the layout of Fig. 2 and new Fig. S9 (former Fig. S7) as suggested.

While producing more scattered locations, the S-P location approach confirms the result of the more accurate correlation based approach, and extends it to more earthquakes.

With respect to the suggested comparison, when we look at the S-P locations considering only the subset of events relocated by waveform correlation (note that S-P times are not available for all 51 events, but for many of them), they are still distributed in a NE-SW band (black circles in Fig. 2b). These events clearly display (Fig. S9) a larger variation of differential times at ALCX (SW), than at ALCN (NW). This means that the distribution of their distances to ALCX must be broader, compared to those to ALCN and, finally (assuming a source depth constraint to a thin layer), that their epicentral distribution must be elongated NE-SW (and – conversely - incompatible with a NW-SE elongation).

3- Moment Tensor Inversion: is there a difference between in Fig.S8 and S10, except that there are more stations? (doesn't need to be changed, I just wonder if I missed something)

In our inversion setup, we combine standard time domain, amplitude spectra (frequency domain) and cross-correlation to best constrain our results.

The difference is that former Fig. S8 (now Fig. S11) shows an example of standard time domain inversion (at close distances), where the misfit to minimize is the L1 norm among observed and synthetic traces, while former Fig. S10 (now Fig. S13) shows an example of cross-correlation based inversion (over a broader distance range), where the misfit to minimize is based on the cross-correlation between observed and synthetics.

4- Depth estimation: l.187 "Measured time delays (...) ARE"

Fig.S11: l.168 "(other events)" ???  can be removed
Corrected.

5- Rupture directivity: l.203 "the estimated rupture velocity"

Fig.5: I didn't understand what the uncertainty mean (blue and red shaded areas in the polar plots)?

Neither how it was obtained, nor how it should be understood?

In the figure caption, l.871, "are plotted"

We corrected the text, as suggested.

We improved the caption of Fig. 5: blue and red shaded areas represent the range of rupture directions, considering uncertainties which are estimated from the residuals. The reference for the uncertainty estimation (López-Comino et al. 2016) has been now cited in the method section.

5- Waveform clustering: as mentioned above, maybe this section could come earlier?

We recognize that there is here some misunderstanding, and therefore improved the text which introduces this method. The purpose of the waveform clustering method (now referred as 'waveform similarity of relocated events' to avoid confusion) is to identify families of events, which share very similar waveforms, simultaneously seen at different stations. This high similarity, ensured at different azimuths and distances, implies a similarity of location and mechanism. This analysis can be performed for relatively large events only, as they are nicely recorded at more stations, and it is interesting when including part of events with accurate locations and/or moment tensor solutions; in such cases, we can infer a similar locations and mechanisms also for the events which are very similar to reference ones.

This is the reason why we chose, and still prefer, to keep this accurate analysis after the sections dedicated to locations and moment tensors.

l.215-216: did not understand the first part of the sentence.

We rephrased the whole paragraph, in line with our answer above.

Again the same comment as previously, but couldn't template matching also define some clusters?

It is true that template matching also provides clusters, as slave events are found to be similar to their masters (templates). In our work, template matching is primarily used to enhance the catalog (relevant for the discussion of the sequence statistics). However, we also use it to better see the migration pattern (where we indeed assign to slave events the same locations of their respective masters). This helps, for example, to better appreciate the migration of seismicity.

As explained above, waveform clustering is used with a slightly different aim, which is to accurately identify similar events among those which are either well localized and/or with a known mechanism. The high similarity is then used to discuss their occurrence on a common fault. We think the revised text is now clearer.

Fig.S13a: could you give some CC values here (or plot the CC matrix, or the silhouette plots)? The waveforms are both similar and (I agree) dissimilar in the details. But just out of curiosity, would be great to know how much they vary.

Fig.S13b: the profiles are a bit difficult to situate/understand...

In view of these comments and also other related comments by other reviewers, we have modified the figure and added some plots. Former Fig. 13a is now included in the main text (new Fig. 6). We included a silhouette plot (as suggested) as new Fig. S16. Finally, former Figure 13b (with a statement on the NE-SW profile in the caption) is prepared as new Fig. S17.

* Discussion:

l.234-235: replace by "such location uncertainties are attributed to (...)"

l.239: "clearly resolve the seismicity": I agree for the 1st relocation method, but less obvious to my eyes for the 2nd method

l.252: change words order  "depth uncertainties were not reported there"

l.275, l.282, l.293: replace "what" by "which" or "that"

l.286: "antithetic": what does it mean in this context? it comes in opposition to what?

l.302: I think there should be a short sentence about the 3rd phase here as well.

All comments above were corrected/considered

l.308: what does "healing backfront" mean here?

Seismicity can temporarily increase in consequence of pore pressure diffusion, which is seen with increasing delay at increasing distances. A secondary healing front has been sometimes observed in other datasets, showing a delayed decrease of the seismicity rates to the original rate. However, since there is no clear evidence of this at Castor, we preferred to remove the sentence.

Fig.6b: did you use the locations of the 411 events here? Maybe only me, but I don't think the temporal migration of the triggering front is that obvious...

We plot here the cross-correlation relocated events (51) and their corresponding slaves (based on the template matching approach), to which we attributed the same locations as their masters, for a total of 964 events. We improved the caption of Figure 7 (former Fig. 6) to make this clear.

We believe a SW migration is clearly visible (see plot Fig. 7c). Since the seismicity is nicely aligned along the NE-SW direction, we decided to highlight the migration in terms of the distance to the injection point (Fig. 7b), which helps the discussion of the underlying processes.

Fig.6d: the scale of the cumulative seismic moment should be adapted! Cannot see anything right now...

We apologize. The panel axes has been corrected to show the temporal evolution of the cumulative seismic moment.

Fig.7: the colours are more difficult to distinguish in this figure. May be a better colour map (otherwise it is OK)? Could you label the Amposta fault in this figure again?

We updated the figure colors in new Fig. 8 (same colors are now also used in some panels of Fig. 2). We labeled the Amposta fault, and made its line thicker.

1.327: b-value from 0.8 to 0.66. Isn't it a substantial change?

After recalculations, we can confirm the change is not substantial (0.77 in phase 3, details below)

Use of 1 vs 2 decimals?

We fixed to 1 decimal, to be consistent (thus it is approximated to 0.8, as in phase 2).

Phase 3 also contains many more events than phase 2. How does it affect the b-value that is a statistical parameter+

The variation does not seem to be substantial. And it is also true that the methods of estimating the b-value in induced anthropogenic sequences could not be valid as the GR is rarely respected.

In our case, however, the increase of events in the catalog follows an exponential trend (especially for $M < 3$) and therefore, we considered appropriate to apply the standard methods for estimating the b-value.

The observed decrease in the b-value, although weak, seems to be a robust feature for different completeness magnitude values of the augmented catalog. The decrease is also found in IGN and EBRO catalogs.

Here, the b-value and its variations are validated for $M_c = 0.5$. The value of M_c for Phase 2 and 3, is evaluated following Herrmann and Marzocchi (2021), that suggests applying in high resolution catalogs the Lilliefors' goodness-of-fit test, able to find inconsistencies in the magnitude distributions. The test applied to the augmented catalog resulted in M_c _Lilliefors close to 0.5.

Finally, we have to admit that the resulting estimates could change when including seismicity before and after the 1-month window for which data are available.

[Herrmann M., and Marzocchi W., 2021. Inconsistencies and Lurking Pitfalls in the Magnitude-Frequency Distribution of High-Resolution Earthquake Catalogs. *Seism. Res. Lett.*, 92, 2A, 909-922. doi: 10.1785/0220200337]

.348: "driveS"

1.352: "phaseS"

1.364: "involve a role"  imply

1.374: "the case FOR"

1.381: "proofing"  proving (also 1.564)

1.384: "more clear"  clearer

All corrected.

General English correction: I noticed some inconsistencies in the use of past/present tenses.

Corrected. All material was carefully checked again by our native speaker coauthor.

Reviewer #2 (Remarks to the Author):

Review of “Seismicity at the Castor gas reservoir driven by pore pressure diffusion and asperities loading” by S. Cesca and others

In this paper, the authors re-analyze and supplement data related to a series of $M > 4$ earthquakes at the Castor injection platform. Given the diversity in previously published results surrounding these earthquakes, this more detailed analysis with additional data is not only warranted, but needed, especially since these earthquakes are the results of induced seismicity. For cases of induced seismicity, it is very important for unambiguous results, if possible, that can be communicated to operators, regulators, and the public. As such, I find this an important study. Further, scientifically this is a very complete seismic analysis that warrants publishing--the suite of tools available for studying this type of sequence has been employed in a systematic and rigorous approach. Overall, the paper is well-written. I make a few specific suggestions for clarity below. I also ask for elaboration in the text describing a couple of the methods.

We are thankful for the careful review and appreciation of our work. We provide replies to all reviewer comments below.

Comments:

The distance geometry (line 459) approach for locating earthquakes is new to me and there are no references.

It would be helpful if the authors could elaborate more on this process and perhaps add a figure to the supplement that shows the process. From what I get, which I am not entirely sure is accurate, you use S-P as a metric to find events spatially close and this largely works because the stations used in the analysis are approximately equidistant from the centroid of the sequence. Then starting from at minimum 4 well-located earthquakes you iteratively add one event at a time and relatively relocate all (?) events based on these S-P times.

We agree that the description of the method was too short, and have enhanced this part of the text. The location method is described in detail in a recently published paper by Grigoli et al. (2021). The method uses the inter-event distance between events pairs as input. The inter-event distance between two events (let us say a and b) is calculated by using the difference of the $T_s - T_p$ between event a and event b at two different stations. The stations do not need to be equidistant. What is most important is that their aperture with respect to the cluster centroid is large enough (generally more than 50 degrees, the optimal condition is 90 degrees). The Castor layout is in this sense optimal, as closest stations ALCN and ALCX are located in almost perpendicular directions from the injection point.

We improved the description of this approach in the Methods section of the manuscript and better referenced the original paper, so that interested readers can get all the mathematical and implementation details there. The method publication was accompanied by software release (HADES). The software is publicly available via gitHUB. We added the link to the software in the manuscript.

[F Grigoli, W L Ellsworth, M Zhang, M Mousavi, S Cesca, C Satriano, G C Beroza, S Wiemer, Relative earthquake location procedure for clustered seismicity with a single station, *Geophysical Journal International*, Volume 225, Issue 1, April 2021, Pages 608–626, <https://doi.org/10.1093/gji/ggaa607>]

Questions:

- If the event is not close to the centroid how do you associate the S-P times between the two stations? The distance (S-P time) will not be the same to each station.

The method is designed to relocate a cluster of events with 1 or 2 stations, in addition to the $T_s - T_p$

times we also use waveform cross-correlation to estimate the “closeness” of a seismic event to the seismicity cluster.

The Ts-Tp time does not need to be the same (or similar at the two stations), this is not required by the method. It is important that the stations have a minimum aperture of ~50 degrees with respect to the centroid of the cluster (the method can also work with smaller apertures but the quality of results decreases because the inter-event distance estimation is affected by larger errors).

The details of the method can be found in Grigoli et al (2021).

- Does the order that you add stations matter? Is there a rule for how close you have to be to one of the 4 well-located events?

The order of the stations does not have any effect. It is interchangeable and produces exactly the same results. There are, in change, some requirements that need to be satisfied for the application of this method. The method has been specifically designed to locate seismicity clusters outside the seismic network, using only one or two stations. The main requirement is that inter-event distances should be small compared to the average source station distance (Grigoli et al. 2021 GJI), which is ensured if source-receiver distances are 2-3 times larger than the maximum elongation of the cluster. This condition can be further relaxed when using two stations (if they satisfy the aperture requirement). In this context, the Castor layout is perfect for the application of the method.

- How different are the results in the absolute locations when using the different velocity models?

The reconstructed cluster shape has a loose dependence on the velocity model, because of its relative location implementation. We performed a sensitivity test by relocating the seismic cluster 15 times with different velocity models. We plot in Fig. S10 the Kernel Density Estimation (KDE) of the (408 x 15) events relocated with all these models (Vp range from 4.0 to 6.0 km/s and Vp/Vs from 1.67 to 1.79). The KDE shows that velocity perturbations produce small variations in the cluster shape.

Adding the array based analysis to get source depth was a great addition! You really needed some independent data to get good depths.

Thank you. We also believe that independent data are key to solve the discussion on the depth, which is indeed challenging using only local data with such an asymmetric geometry. Using array methods is indeed a potential approach for all those cases, where the lack/limitations of local data challenge the depth assessment.

Template matching.

- In Fig S3, there are several stations where the correlations do not look all that good. These do tend to correspond to low-correlation values, but this begs the questions, what is the threshold that you used to declare a detection?

- You also seem to match on the S-wave and from Fig. S3 the P-wave arrival is not all that obvious. Were you able to get P-picks for the relocations?

- Are the blue lines in Fig S3 the theoretical S-wave arrival or the start of the data window?

The threshold used is set to 15 times the median absolute deviation (MAD) of the stacked cross-correlation function. The stack of cross-correlations requires at least 8 channels.

New detections are co-located as templates and used for analyzing the spatiotemporal pattern.

Relocated events represent only a small subset, for which good picks are available.

The blue vertical lines as specified in the new Figs. S3-S5 represent theoretical S-wave phase arrivals. The theoretical S-phase arrivals are used as a reference to trim the templates, and does not affect the template matching performance if the template window includes part of the seismic event and does not consider noise.

b-values: In Fig. S5, the fit to the injection catalog is not good for $M > 1.5$. In fact, the fit is only good over 1 magnitude unit, so I do not find this b-value meaningful. My guess is that the M_c value is artificially low perhaps as a result of using magnitudes from the template matching. If you choose

to persist with the argument of a b-value difference this section needs additional work and you need to show a fit over the entire magnitude range.

In complex seismic sequences such as those induced, where earthquakes are often organized in swarms or foreshocks before triggering events with a maximum magnitude, the evaluation of b-values for short time windows can be questionable and lead to abnormally low values. It can happen by using template matching that the frequency-magnitude distribution does not follow the Gutenberg-Richter (GR) law. The breakdown of the GR law can be caused by spatial heterogeneity of detection capability and the nature and magnitude distribution of swarms. Here, the augmented catalog, obtained by template matching, appears to follow the GR exponential distribution of magnitudes (see Figure below), validating the variation of b-value consequently. At Castor, a change in b-value has already been discussed in previous publications. Our findings confirm a change of earthquake statistics in mid-September.

The b-value and its variations are also validated for $M_c < 1$. The value of M_c is estimated following Herrmann and Marzocchi (2021) that suggests to apply in high resolution catalogs the Lilliefors' goodness-of-fit test able to find inconsistencies in the magnitude distributions. The test applied to the augmented catalog resulted in $M_c_Lilliefors = 0.5$.

[Herrmann M., and Marzocchi W., 2021. Inconsistencies and Lurking Pitfalls in the Magnitude-Frequency Distribution of High-Resolution Earthquake Catalogs. *Seism. Res. Lett.*, 92, 2A, 909-922. doi: 10.1785/0220200337]

Amplitude spectra (Fig S9): Most of the fits look really good. I was curious about the middle column transverse and radial component, which seem to have a much larger misfit than the other fits. Is this a function of the velocity model or the source properties?

The quality of the fit is, in general, very good. For a few traces, such as the transversal component at station ETOB, there is some larger deviation. We checked that similar fits are found when using the other tested velocity models, so that we would not attribute this feature to a specific velocity model. A source effect is also not very likely, since the fit at other stations with similar azimuth is very good. In conclusion, and given that this minor mismatch is seen only on horizontal components, we can hypothesized that this may be due to some small sensor misorientation.

Line 129: it would be helpful to add “(detailed below in Methods section)” after techniques. Given the reference to the supplementary figures, it was not straightforward to me where to find the details regarding the processing, especially with the non-standard format (methods at the end).

Done. We have to follow the journal format, with methods at the end.

Line 172:, please add a reference after inversion approach. It is referenced in the methods section, but it would be helpful to have it referenced in the results section as well.

Done.

Fig S4, since there is overlap in the two catalogs, perhaps make the Erbo catalog open symbols, so that extended catalog can be better seen

Done. We have improved the layout of Fig. S6 (former Fig. S4), increasing symbol sizes and adding high symbol transparency to better see the two catalogs.

Reviewer #3 (Remarks to the Author):

General Comments

This a very thorough and detailed manuscript about induced seismicity at the Castor gas reservoir offshore Spain proved to be caused by gas storage injection, which is considered rare. Because of previous work showing mixed conclusions, this study presents some detailed seismological analysis using new approaches that quite frankly would be difficult to improve upon. This manuscript is well written with clear figures, and I have few issues with any part of the manuscript. My main concern is that not enough detail is provided on the techniques used to be able to reproduce the results. The science results are very interesting and important, and overall, this is an excellent paper.

Thank you for your positive feedback and valuable comments. We improved the paper to make sure all our results can be reproduced. We reply to single comments below.

Specific Comments

Some of the detail to the techniques are either not presented, are after the results, and or are in the supplement. I do not know if this is a constraint of the requirements of the manuscript, but I would suggest have the methods sections before results. When I first started the template matching description, and I has no idea how this was done, and then realized after getting through all of re results and discussion, this was described in the methods section at the end of the paper. If this is to remain the structure of the paper, it would be good to add a sentence or to describe in words the approaches.

We improved the manuscript and supplement to ensure that full details are provided on all used techniques. Unfortunately, we cannot modify the paper structure as in the first reviewer's suggestion, as we have to respect the journal layout. We try to go for the second suggestion, adding a short paragraph on the approaches when introducing our results, and making clear where further informations can be found.

Lines 152-154: "used in previous papers." What previous papers? Need to cite them here. Again, this is with details to the approach. I have no idea what "advanced relative earthquake localization methods" are being used. (I found it in the methods section... too late for me... confused me to no end).

We include proper citations.

Lines 171-173. What probabilistic approach was used? Not cited here. How do we know what is "extremely stable"? We have no context if this is standard for the approach. I understand that the choice of models does not impact result, making it stable. However, the order of the wording is critical here.

We reformulated the text (and avoid using "extremely" stable, which cannot be easily quantified) and provide more information on the probabilistic inversion concept. This is explained in great details in the referenced publications.

Line 184-192. As mentioned in the text, the geometry of the array and the lack of being within a focal depth of the events makes it very difficult to determine depths. I appreciate the effort here to use depth phases. However, I have yet to see an approach for local waveforms that convinces that you are modelling depth phases (perhaps I just have not read the right papers). Green's functions are difficult to calculate at local distances because of local geology complexities. Not having the details of the approach (how synthetic waveforms were calculated) makes this section even less convincing.

The depth analysis is performed with independent data, using stations at far-regional and teleseismic distances.

We clarified this issue and extended the methodological discussion for the array depth modeling (in short in the method section, where we have length constraints, and more in detail in the supplementary material). Over these distances, Green's functions are computed using a reflectivity approach (QSEIS), and considering a local crustal model at the source (here model G), a local crustal model at the array location (here, extracted from the CRUST2.0 database) and a common mantle model (AK135). The Green functions with this setup can be computed for bodywaves only. The shallow part of the local model is important as it controls how the differential pP-P time is mapped into a depth, therefore we test also alternative models.

In our modeling, we assume a range of possible source depth, and we look at P pulses and consequent depth phases. Finally, since the focal mechanism is known (from the moment tensor inversion), we can compute synthetic beams for the proper mechanism and different depths and compare them to the observed beam.

The algorithm itself is open source and (we reference now the GitHub repository).

The method is suited for relatively shallow seismicity. Major limitations are related to the magnitude and depths. For weak earthquakes, the signal may have a poor quality at distant array. In our case, for example, the signal is clear e.g. at GERES, but less clear at other arrays at too large distance. As for the depth, this can be best resolved when the pP-P delay is larger than the duration of the rupture: e.g. for larger earthquake, with a longer rupture duration, the pP and P pulses may overlap, challenging the beam modeling. At Castor we can see the two pulses separated, given that the rupture durations do not exceed 0.5 s, and the depth phase delays are clearly larger.

Finally, the method has been successfully applied in a number of recent applications (see a number of references below):

Negi et al. 2017, Tectonophysics, <https://doi.org/10.1016/j.tecto.2017.05.007>

Gaebler et al., 2019, Solid Earth, <https://doi.org/10.5194/se-10-59-2019>

Mouslopoulou et al., 2020, Geochemistry Geophysics Geosystems (G3), <https://doi.org/10.1029/2020GC009243>

Büyükkapınar et al., 2021. Frontiers in Earth Science, <https://doi.org/10.3389/feart.2021.663385>

Lines 212-227: Figures of this analysis are being presented only in supplemental material. Is it worth presenting if it is only supplemental?

The figure is actually very important for the slightly updated discussion. So, we decided to move part of it to the main document.

Lines 230-267: Outstanding discussion on depth resolution.

We are not sure what is here suggested. We changed a few words to make clear that the depth is now accurately resolved using independent data (i.e. using seismic arrays and a dedicated method), as also pointed out by other reviewers.

Line 275: "what excludes", should it be "which excludes"?

Correct.

The methods section should be moved up before results. I find it very difficult to understand the results without knowing what was done exactly (so as to believe the results).

See our reply above.

REVIEWERS' COMMENTS

Reviewer #1 (Remarks to the Author):

I read both the reply to reviewers and the corrected manuscript and I thank the authors for taking into account the comments and for clarifying a few points I missed in the first round. Newly added material is also useful. It is good that the method section is now referred early enough, so that the reader is not overwhelmed by the different and very specific techniques that have been applied before discovering that their description comes later.

I do not have any further comments and I think the manuscript is ready for publication.

Reviewer #2 (Remarks to the Author):

The authors did a nice job addressing the comments from my first review. I have a couple of very minor editorial comments from this read, but overall this is a very nice and comprehensive analysis of the seismic sequence at the Castor gas reservoir.

The text in lines 113-114 reads "Unfortunately, one of the possible two plane orientations fit one of the proposed seismicity distributions, so the fault geometry remains unresolved." I think there is a typo and this should read "Unfortunately, none"

The clause "On the contrary, they" connecting the sentences in lines 222 and 223 is awkward. I recommend replacing this with "and"

Please provide a reference or an equation for the silhouette score (line 610).

--Kristine Pankow

Reviewer #3 (Remarks to the Author):

General Comments

I reviewed the previous version and have reviewed the response to reviewers, which included my comments. This version of the manuscript is much improved and will be a solid contribution to the field of induced seismicity. The paper reads really well, with the results smoothly referring to the methods section, which was problematic in the previous version. I have no substantive comments, just a few minor edits (see Specific Comments)

Specific Comments

Line 163, 254, 591, I never like the use of "clearly" in "clearly resolve" or "clearly identified: Clearly is objective. Remove this sort of modifier from the text.

Line 96: "...away, The..." should be "...away. The..."

Line 171: "orientationof" should be "orientation of"

Reviewer #1 (Remarks to the Author):

I read both the reply to reviewers and the corrected manuscript and I thank the authors for taking into account the comments and for clarifying a few points I missed in the first round. Newly added material is also useful. It is good that the method section is now referred early enough, so that the reader is not overwhelmed by the different and very specific techniques that have been applied before discovering that their description comes later.

I do not have any further comments and I think the manuscript is ready for publication.

Thanks for the review.

Reviewer #2 (Remarks to the Author):

The authors did a nice job addressing the comments from my first review. I have a couple of very minor editorial comments from this read, but overall this is a very nice and comprehensive analysis of the seismic sequence at the Castor gas reservoir.

Thanks for the positive review. We considered all remaining minor comments.

The text in lines 113-114 reads "Unfortunately, one of the possible two plane orientations fit one of the proposed seismicity distributions, so the fault geometry remains unresolved." I think there is a typo and this should read "Unfortunately, none"

We rephrased the sentence as: "Unfortunately, each of the possible two plane orientations fits one of the proposed seismicity distributions, so the fault geometry remains unresolved."

The clause "On the contrary, they" connecting the sentences in lines 222 and 223 is awkward. I recommend replacing this with "and"

Corrected as suggested.

Please provide a reference or an equation for the silhouette score (line 610).

We added a reference.

Reviewer #3 (Remarks to the Author):**General Comments**

I reviewed the previous version and have reviewed the response to reviewers, which included my comments. This version of the manuscript is much improved and will be a solid contribution to the field of induced seismicity. The paper reads really well, with the results smoothly referring to the methods section, which was problematic in the previous version. I have no substantive comments, just a few minor edits (see Specific Comments)

Thanks for the appreciation to our work. We considered all remaining edits.

Specific Comments

Line 163, 254, 591, I never like the use of "clearly" in "clearly resolve" or "clearly identified: Clearly is objective. Remove this sort of modifier from the text.

We removed the wording 'clearly' as suggested

Line 96: "...away, The..." should be "...away. The..."

Line 171: "orientationof" should be "orientation of"

Corrected